



**Ensembling Differentiable Process-based and Data-driven Models with**
**Diverse Meteorological Forcing Datasets to Advance Streamflow**
**Simulation**
Peijun Li[1], Yalan Song[1], Ming Pan[2], Kathryn Lawson[1], Chaopeng Shen[1]
[1]Civil and Environmental Engineering, The Pennsylvania State University, University Park,
PA, USA
[2]Center for Western Weather and Water Extremes, Scripps Institution of Oceanography,
University of California San Diego, La Jolla, CA, USA
*Correspondence to: Peijun Li, pql5336@psu.edu; Chaopeng Shen, cshen@engr.psu.edu
**Abstract**
Streamflow simulations via different hydrological models have different features and
can provide valuable information after being ensembled. While few studies have focused on
ensembling simulations via models with significant structural differences and evaluating
them under both temporal and spatial tests. Here we systematically evaluated and utilized the
simulations from two highly different models with great performances: a purely data-driven
long short-term memory (LSTM) network and a physics-informed machine learning
("differentiable") HBV (Hydrologiska Byråns Vattenavdelning) model ($\delta$HBV). To
effectively display the features of the two models, multiple forcing datasets are employed and
utilized in two ways. The results show that the simulations of LSTM and $\delta$HBV have distinct
features and complement each other well, leading to better Nash-Sutcliffe model efficiency
coefficients (NSE) and improved high-flow and low-flow metrics across all spatiotemporal
tests, compared to within-class ensembles. Ensembling models trained on a single forcing
outperformed a single model using fused forcings, challenging the paradigm of feeding all
available data into a single data-driven model. Most notably, $\delta$HBV significantly enhanced



spatial interpolation when incorporated into LSTM, and provided even more prominent
benefits for spatial extrapolation where the LSTM-only ensembles degraded significantly,
attesting to the value of the structural constraints in δHBV. These advances set new
benchmark records on the well-known CAMELS (Catchment Attributes and Meteorology for
Large-sample Studies) hydrological dataset, reaching median NSE values of ~0.83 for the
temporal test (densely trained scenario), ~0.79 for the ungauged basin test (PUB, Prediction
in Ungauged Basins), and ~0.70 for the ungauged region test (PUR, Prediction in Ungauged
Regions). This study advances our understanding of how various model types, each with
distinct mechanisms, can be effectively leveraged alongside multi-source datasets across
diverse scenarios.





**Highlights**

- Combining LSTM and δHBV with diverse forcings sets new accuracy benchmarks

- Ensembling models with one forcing outperforms merging forcings as an input

- δHBV and LSTM together always increase NSEs, especially spatial generalization

- δHBV provides valuable spatial constraints in the deterministic ensemble simulations

- δHBV and LSTM have different error characteristics that can be offset in an ensemble

**Keywords**

Streamflow simulation, differentiable model, deep learning, hybrid modeling, multi-source
fusion

**1. Introduction**

Streamflow, a critical component of the global hydrosphere, profoundly influences both
human society and natural ecosystems (Lins and Slack, 1999). Accurate simulation and
prediction of streamflow yield numerous benefits, including improved flood prevention
strategies (Brunner et al., 2021). Hydrological models serve as indispensable tools for
achieving this objective and can be traditionally categorized into two types: data-driven
models (Feng et al., 2020; Kratzert et al., 2018; Liu et al., 2024; Nearing et al., 2024) and
process-based (or physically-based) models (Newman et al., 2017; Paul et al., 2021).
Data-driven models, exemplified by long short-term memory (LSTM) (Feng et al., 2020;
Kratzert et al., 2018) and transformer (Liu et al., 2024) networks, excel in learning patterns
from multi-source data (Li et al., 2023b, 2024; Liu et al., 2022; Nearing et al., 2024) and
generally achieve high performance. However, they often lack interpretability and may not
resolve extreme values very well (Li et al., 2020a; Song et al., 2024b). Conversely,



process-based models, derived deductively from physical laws or conceptualized views of
natural systems, offer insights into internal hydrological processes but may exhibit weaker
performance due to structural inadequacies (Li et al., 2020a; Zhang et al., 2019).
To combine the benefits and counteract the weaknesses of these two kinds of models,
many efforts have been made to incorporate physical constraints and structures into
data-driven models to align with fundamental physical principles, such as mass and water
balances (Bandai and Ghezzehei, 2021; Wang et al., 2020; Xie et al., 2021). The most
seamless integration uses neural networks to provide parameterizations or missing process
representations for process-based models (Aboelyazeed et al., 2023; Bindas et al., 2024; Feng
et al., 2022; Jiang et al., 2020; Kraft et al., 2022; Rahmani et al., 2023; Song et al., 2024c;
Tsai et al., 2021). These differentiable models (Shen et al., 2023) connect (flexible amounts
of) prior physical knowledge to neural networks, and have displayed many advantages,
including improved computational efficiency and prediction of untrained variables (Tsai et
al., 2021), spatial generalization (Feng et al., 2023b), and representation of extremes (Song et
al., 2024b). However, it is also unclear whether current differentiable models, e.g., δHBV, the
Hydrologiska Byråns Vattenbalansavdelning (HBV) model implemented within a
differentiable framework  (Feng et al., 2023b; Shen et al., 2023; Song et al., 2024b), have
unique bias characteristics that are associated with the process-based parts of their structures
that cannot be reduced once the equations are prescribed.
Orthogonal to such efforts are ensemble simulations (Yu et al., 2024), which combine
many members with different biases and uncertainties to mitigate their respective biases in
deterministic predictions. Many previous studies have tried ensemble methods to improve
streamflow (Clark et al., 2016; Zounemat-Kermani et al., 2021) based on many factors, like
initial conditions (e.g., initial weights and biases in LSTM (Kratzert et al., 2018)), data used
for parameterization (Feng et al., 2021), and objective functions (Lin et al., 2024). These



studies generally use one model to generate the differences among the ensemble members.
Furthermore, some studies (Dion et al., 2021; Solanki et al., 2025) have utilized simulations
from multiple different models but are limited to process-based models and resulted in
ensemble simulations that are better than each individual member. Thus far, however, most
studies focus on the simulations from only similar models or model types, and little work has
tested an ensemble across the boundary of model types, especially between data-driven,
process-based, and hybrid models, especially on a large number of samples. Presumably if
each model has its own unique bias, data-driven and process-based models are likely to
exhibit greater differences due to their inherently distinct characteristics. It remains unclear
whether ensembling across model types should bring benefits to deterministic predictions.
Furthermore, grounded in the process-based model, the differentiable process-based
hydrological model, such as δHBV, significantly enhances performance compared to
traditional process-based models, while on the other hand introducing greater uncertainty
regarding its potential benefits when ensembled. Moreover, previous studies have primarily
focused on evaluating ensemble simulations for temporal predictions. However, streamflow
simulation under spatial extrapolation scenarios presents greater challenges, and findings
from temporal tests may not be directly applicable in this context.
It is known that the performance of any type of hydrologic model heavily depends on
the quality of input data, particularly meteorological forcing data (Bell and Moore, 2000; Yao
et al., 2020), and other inputs like the uncertainties of initial conditions can be mitigated via
warming up (Yu et al., 2019). While independent forcing datasets excel in certain aspects,
they each carry different error characteristics (Beck et al., 2017; Behnke et al., 2016;
Newman et al., 2019) and accordingly affect the hydrological models in different ways. In
order to fully display the different features between LSTM and δHBV, multiple forcing
datasets could be considered. Given the utilization of multiple forcing datasets, one could



choose to use data fusion to combine them into a single coherent model input (Kratzert et al.,
2021; Sawadekar et al., 2024), or to pass each forcing dataset through a model and then
afterwards combine the multiple outputs in an ensemble. It is not clear which approach is
more beneficial.
Considering the knowledge gaps discussed above, we sought to answer several research
questions:
1. Will a cross-model-type ensemble of LSTM and δHBV improve deterministic

streamflow prediction more than a within-class ensemble?

2. Is it better to use multiple forcings in one model or to ensemble multiple models, each

with a different forcing input?

3. Do process-based equations bring unique value to an ensemble, especially in terms of

spatial generalizability?

The remainder of this paper is structured as follows: Sect. 2 outlines the hydrological
data and models used in this study, as well as the experimental design. Results and
discussions are presented in Sect. 3, with conclusions provided in Sect. 4.

## 2. Material and Methods

2.1. The CAMELS hydrologic dataset
The Catchment Attributes and Meteorology for Large-sample Studies (CAMELS)
dataset (Addor et al., 2017) is widely employed for hydrological model evaluation and
community benchmarking. The CAMELS dataset encompasses 671 basins distributed across
the conterminous United States, with basin sizes ranging from 1 to 25,800 km² (median: 335
km²). This standardized and publicly available dataset serves as a benchmark for evaluating
various hydrological models, with LSTM models trained on this dataset often serving as a
reference point for comparing other models (Kratzert et al., 2021). CAMELS provides



basin-scale data, including streamflow observations and static basin attributes, as well as
forcing datasets from three independent sources: Daymet (Thornton et al., 1997), North
American Land Data Assimilation System (NLDAS) (Xia et al., 2012), and Maurer (Maurer
et al., 2002). Each of the three meteorological forcing datasets operates at a daily temporal
resolution, encompassing precipitation, temperature, vapor pressure, and surface radiation
variables, with daily temperature extrema of NLDAS and Maurer supplemented from
Kratzert et al. (2021). These three meteorological forcing datasets have methodological
distinctions in spatial resolution, data generation approaches, and temporal processing
(Behnke et al., 2016; Kratzert et al., 2021). Exemplary plots illustrating the differences
among the three meteorological forcing datasets are provided in Appendix B. These features
can lead to dataset-specific error characteristics and make them valuable for displaying the
distinct features of different model types. All model inputs used in this study are detailed in
Table C1.

2.2. Long short-term memory
As one kind of deep learning algorithm, long short-term memory (LSTM) (Hochreiter
and Schmidhuber, 1997) has unique structures like hidden states and gates activated by the
tanh and sigmoid functions (Li et al., 2023a), respectively. These features enable LSTM to
excel in streamflow simulation tasks (Feng et al., 2020; Kratzert et al., 2018; Nearing et al.,
2024). In the current benchmark framework, LSTM models are trained using dynamic
atmospheric forcings and static basin attributes as inputs, with streamflow as the target
output, making it perform well in both temporal and spatial tests (Figure 1a). In this work, for
cross-group comparability, we used the LSTM model and its hyperparameters as reported in
Kratzert et al. (2021).



2.3. Differentiable HBV model (δHBV)
The Hydrologiska Byråns Vattenbalansavdelning (HBV) model is a parsimonious
bucket-type hydrologic model that simulates various hydrological variables, including snow
water equivalent, soil water, groundwater storage, evapotranspiration, quick flow, baseflow,
and total streamflow (Aghakouchak and Habib, 2010; Beck et al., 2020; Bergström, 1976,
1992). Recently demonstrated differentiable HBV (δHBV) model (Feng et al., 2023b; Shen et
al., 2023; Song et al., 2024c) incorporates deep neural networks for both regionalized
parameterization and missing process representations within a differentiable programming
framework that supports "end-to-end" training (Figure 1b). This innovation enables δHBV to
effectively learn from data while obeying physical laws, resulting in high-level performance
for streamflow simulations. From the perspective of process-based modeling, LSTM is a
regionalized parameter provider that leverages the autocorrelated nature of its inputs to
impose an implicit spatial constraint on the generated parameters.
In this study, we used δHBV1.1p (Song et al., 2024c, b) which is an updated version
from δHBV1.0 (Feng et al., 2022, 2023b). The main improvement is the addition of a
capillary rise module, which enhances the characterization of low flows. Other modifications
include the use of three dynamic parameters during the warm-up, training, and test periods,
the removal of log-transform normalization for precipitation, and the adoption of NSE as the
loss function for model training. The basic equations in δHBV are as follows:

$$\theta = LSTM_w(\bar{x}, \bar{A}_{attr}) \tag{1}$$

$$Q = HBV(x, \theta) \tag{2}$$

$$W_{opt} = argmin_w(L(Q, Q^*)) \tag{3}$$

where $\theta$ are the dynamic or static physical parameters, $w$ denotes the weights and biases of
LSTM, $x$ includes the basin-averaged meteorological forcings, such as precipitation, mean



temperature, and potential evapotranspiration, with $\bar{x}$ representing their normalized versions.
Similarly, $\bar{A}_{attr}$ consists of normalized observable basin-averaged attributes, encompassing
basin area, topography, climate, soil texture, land cover, and geology (Table C1).
Precipitation and mean temperature are from CAMELS, while potential evapotranspiration is
calculated based on the Hargreaves (1994) method using mean, maximum, and minimum
temperatures along with basin latitudes, all from data described in sect. 2.1. $Q$ and $Q^*$ are the
streamflow simulations (model outputs) and observations (as provided in CAMELS),
respectively. HBV is implemented on PyTorch so it is programmatically differentiable: all
steps store information related to gradient calculations during backpropagation, allowing this
model to be trained together with neural networks in an end-to-end fashion. More details
about differentiable HBV can be found in previous studies (Feng et al., 2022; Song et al.,
2024c). The details of some particularly relevant HBV processes are described in Appendix
A.

2.4. Experimental Design
In this study, we trained the two models in highly different types (LSTM and δHBV)
using three meteorological forcing datasets (Daymet, NLDAS, and Maurer), resulting in six
corresponding streamflow simulations (Figure 1c) for each different test scenario (see sect.
2.5 for additional information). The training processes of LSTM and δHBV followed Kratzert
et al (2021) and Feng et al. (2023b), respectively. Test results and performance metrics for all
models are reported for the 531-basin subset that excludes those with areas larger than 2,000
km² or with more than a 10% discrepancy between different basin area calculation methods
(Newman et al., 2017).
To generate ensembles, we tested various weighting strategies and ultimately employed
averaging to combine the six single-forcing, single-model-type simulations, as it yielded the



best performance. To better describe various combinations including cross-model ensembles,
these simulations were categorized into six groups (Table 1). A shorthand notation is used
throughout the remainder of this work to describe the forcing datasets and ensembles.
Daymet, NLDAS, and Maurer are abbreviated as superscripts 1, 2, and 3, respectively. The +
symbol is used to group model types being ensembled, while superscript clustering (e.g., $^{12}$ or
$^{123}$) is used to group the meteorological forcing types being ensembled, with parentheses
indicating that the superscripts apply to all model types within. For example,
$(LSTM + \delta HBV)^{123}$ could be explicitly written as
$LSTM^1 + LSTM^2 + LSTM^3 + \delta HBV^1 + \delta HBV^2 + \delta HBV^3$. To compare two different
strategies to utilize the multiple meteorological forcing datasets and to benchmark against the
previously highest performance, we additionally trained a single LSTM model using all three
forcing datasets as simultaneous inputs as done by Kratzert et al. (2021), referred to as
LSTM$^{\text{multi}}$ (the last row in Table 1).



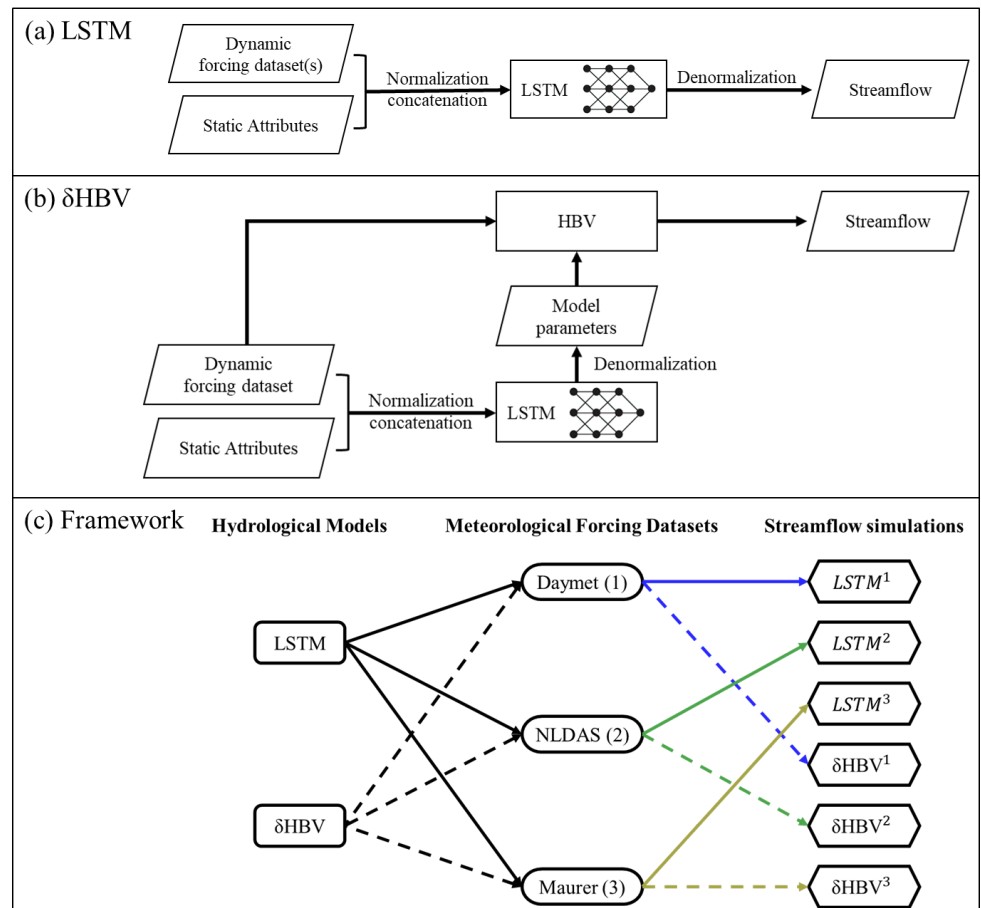

*Figure 1. (a) The LSTM structure, (b) the δHBV structure, and (c) the framework to generate the six individual ensemble members of the streamflow simulations, in which different colors of arrow lines denote the different meteorological forcing datasets (also denoted as 1, 2, 3) and the arrow line styles (solid and dashed lines) indicate the LSTM and δHBV models, respectively.*





*Table 1. (a) The six groups of streamflow simulations, and (b) the streamflow simulation via*
*LSTM based on a different strategy to utilize three meteorological forcing datasets (Kratzert*
*et al., 2021). Superscripts 1, 2, and 3 denote Daymet, NLDAS, and Maurer, respectively. The*
*ensemble across forcings ("ef") superscript indicates an ensemble of model simulations, each*
*of which uses a different single meteorological forcing, e.g., $LSTM^{12}$ means the average of*
*$LSTM^1$ and $LSTM^2$.*

| (a) Six Groups of Streamflow Simulations | |
|---|---|
| **Group Name** | **Group Members** |
| LSTM | $LSTM^1$, $LSTM^2$, $LSTM^3$ |
| $\delta$HBV | $\delta HBV^1$, $\delta HBV^2$, $\delta HBV^3$ |
| LSTM+$\delta$HBV | $(LSTM+\delta HBV)^1$, $(LSTM+\delta HBV)^2$, $(LSTM+\delta HBV)^3$ |
| $LSTM^{ef}$ | $LSTM^{12}$, $LSTM^{13}$, $LSTM^{23}$, $LSTM^{123}$ |
| $\delta HBV^{ef}$ | $\delta HBV^{12}$, $\delta HBV^{13}$, $\delta HBV^{23}$, $\delta HBV^{123}$, |
| $(LSTM+\delta HBV)^{ef}$ | $(LSTM+\delta HBV)^{12}$, $(LSTM+\delta HBV)^{13}$, $(LSTM+\delta HBV)^{23}$, $(LSTM+\delta HBV)^{123}$ |

| (b) Using forcing datasets as simultaneous inputs to an LSTM | | |
|---|---|---|
| **Streamflow Simulation** | **Model Type** | **Meteorological Forcing Dataset** |
| $LSTM^{multi}$ | LSTM | Daymet, NLDAS, Maurer |


2.5. Evaluation Scenarios and Criteria
The above cases were comprehensively evaluated for performance in temporal
extrapolation (Feng et al., 2022; Kratzert et al., 2018), as well as two types of spatial
generalization: prediction in ungauged basins (PUB) (Feng et al., 2023b; Kratzert et al.,
2019), and prediction in ungauged regions (PUR) (Feng et al., 2021, 2023b):
● **Temporal Test**: Models were trained using data from all basins and tested across





different periods.

- **PUB Test**: Models were trained on randomly selected subsets from all basins and tested on the remaining basins during the same time period.

- **PUR Test**: Different from the PUB test, basins were grouped into continuous regions, one of which was selected to comprise the group of testing basins while the others were used for training.

Temporal generalization is generally considered to be the easiest of these tests. In terms of spatial generalization, which approximates data-sparse scenarios, the PUB test is an example of spatial interpolation, whereas the PUR test involves spatial extrapolation. The PUR test is widely regarded as the most challenging and may therefore produce findings that differ significantly from those in other scenarios. In this study, all basins were divided into 10 groups for the PUB test and 7 groups for the PUR test (Table 2) in the same way as Feng et al. (2023b). The spatial extent of 7 regions for PUR test is also shown in Figure 3(c1-c2). Therefore, we conducted 10 rounds for the PUB test and 7 rounds for the PUR test, with a different group held out for testing in each round. Model performance was evaluated after concatenating the test results for all basins.

*Table 2. Differences of temporal, PUB, and PUR tests.*

| Test Scenario | Training | | Testing | |
|---|---|---|---|---|
| | Basin | Time | Basin | Time |
| Temporal | All[a] | 1980-1995[b] | All | 1995-2010 |
| PUB | Random nine-tenths | 1980-1999 | Holdout[c] | 1995-1999 |
| PUR | Random six of seven regions | 1980-1999 | Holdout | 1995-1999 |

[a]*δHBV training followed Feng et al. (2023b) using all 671 CAMELS basins, while LSTM training followed Kratzert et al (2021) using the selected 531-basin subset. Test results and*



*performance metrics for all models are reported for the 531 basins.*
*[b]Each hydrological year spans from October 1st to September 30th of the following year.*
*[c]In the PUB and PUR tests, models are run for 10 and 7 rounds, respectively, with the group*
*held out for testing changed in each round. The simulation performance was evaluated after*
*concatenating the test results for all basins.*

265  We repeated all the simulations with three different random seeds. Therefore, all the

simulations come from a total of $(2\times3+1)\times(1+10+7)\times3$ trained models. The first factor
represents the models: two model types (LSTM and δHBV) trained separately with each of
the three forcing datasets, along with $LSTM^{multi}$, a single model instance trained using all
three forcing datasets simultaneously. The second factor accounts for the three types of tests
(temporal, PUB, and PUR tests), and the last for the three random seeds. With respect to
random seeds, we present two variations in the results, which are visually depicted in Figure
C1. The results without "seed" as a subscript represent the average metric values from
multiple streamflow simulations, each generated from a single model implementation, along
with the corresponding uncertainties, visualized using error bars. The results marked with
"seed" as a subscript are the average of multiple streamflow simulations conducted with
different random seeds. In terms of computational cost, training LSTM and δHBV for
temporal testing under a single meteorological forcing dataset takes approximately 5 and 21
hours, respectively, using a single NVIDIA Tesla V100 GPU.

279  We calculated several well-established performance metrics: Nash-Sutcliffe model

efficiency coefficient (*NSE*) (Nash and Sutcliffe, 1970), Kling-Gupta model efficiency
coefficient (*KGE*) (Kling et al., 2012), percent bias (*PBIAS*), and root-mean-square error
(*RMSE*). We also considered *RMSE* values for high (top 2% "peak" flow, *highRMSE*), low
(bottom 30% "low" flow, *lowRMSE*), and mid-range (the remaining flow, *midRMSE*) flow
conditions (Yilmaz et al., 2008). These metrics were computed for each basin and aggregated
into error bars and cumulative density functions (CDFs). Detailed descriptions of these



metrics and their calculations are available in Table C2. For brevity, the main text primarily
reports NSE values, and other metric values are provided in Appendixes D and E.

**3. Results and Discussion**
3.1. Temporal extrapolation
For the temporal test, in which models were trained and tested on the same basins but in
different time periods, we found that cross-model-type ensembles noticeably surpassed the
within-class ensembles when other conditions were the same with small uncertainties (shown
by the error bars in Figure 2). With a single forcing dataset, the median NSE was elevated
from ~0.735 for LSTM to ~0.79 with δHBV added, though δHBV performance was similar
to LSTM (~0.74 under Daymet). Even after LSTM achieved very high performance when its
simulations, each derived separately from different meteorological forcing datasets, were
ensembled (ef = 123, ~0.808), adding δHBV still improved the results to ~0.818. This finding
was robust for all different combinations of the tested meteorological forcing datasets.
Conversely, adding LSTM also helped to improve δHBV ensembles. These results highlight
the benefits of the cross-model-type ensemble framework, and indicate distinct simulation
features via each model type. LSTM is a data-driven method that has low bias and large
variance. Errors with data (Li et al., 2020b), different sampling strategies (Nai et al., 2024), or
even different weight initializations (Narkhede et al., 2022) can lead to substantively different
outcomes. On the other hand, δHBV may have a smaller variance but a larger bias due to the
fixed HBV formulation (Moges et al., 2016) for some scenarios like low flows (Feng et al.,
2023b; Song et al., 2024c) or in basins with significant water uses (Song et al., 2024a). These
errors with varying characteristics from different model classes can partially offset each other
in an ensemble. On a side note, δHBV models seem more reliant on the quality of the forcing
data as shown in Figure 2. δHBV with the Maurer and NLDAS forcing datasets generally





performs worse than with Daymet that has lower biases. However, even in those cases, the
combination of LSTM and δHBV was still better than LSTM alone, attesting to the
robustness of these benefits.


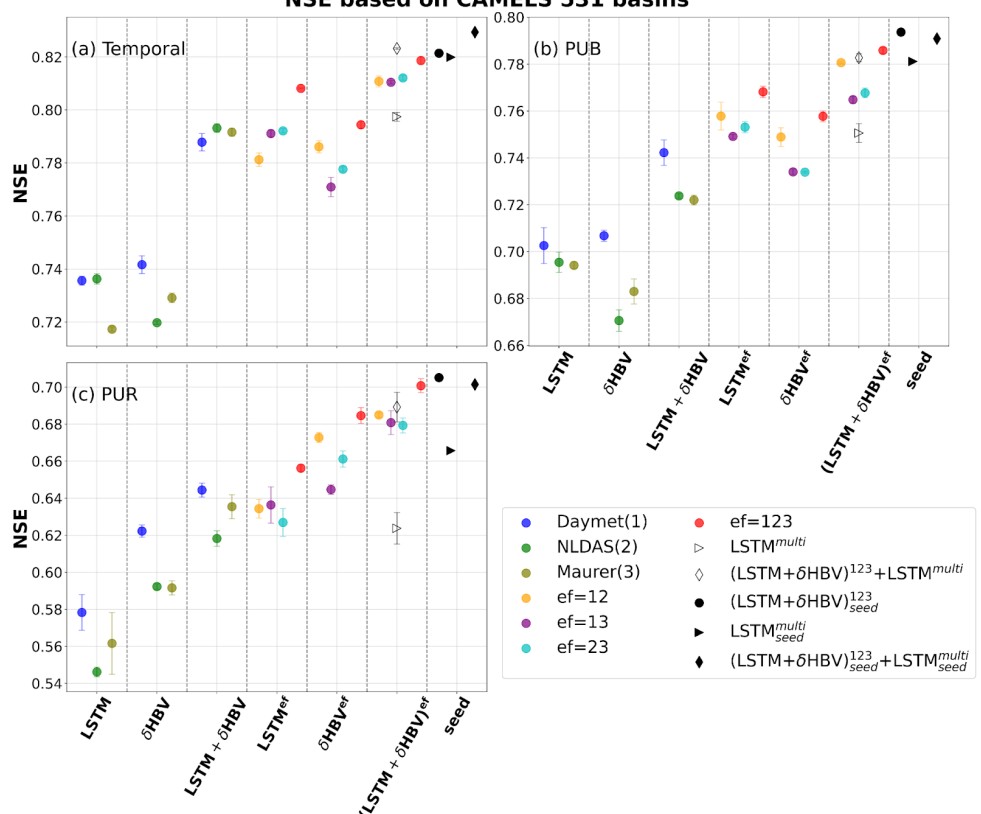


*Figure 2. Median NSE values for 531 CAMELS basins, indicating model and ensemble*
*performances for (a) temporal, (b) prediction in ungauged basin (PUB), and (c) prediction in*
*ungauged region (PUR) tests. Different simulations are represented by variously-shaped and*
*-colored points, and are organized by ensemble group, listed along the x-axis: LSTM, δHBV,*
*LSTM+δHBV, and their "ensemble forcing" counterparts, $LSTM^{ef}$, $δHBV^{ef}$, and*
*$(LSTM + δHBV)^{ef}$. $LSTM^{multi}$ is a single LSTM model trained directly on all three forcing*
*datasets at once. The superscript "ef" denotes the forcing datasets involved in each ensemble*
*(choices of 1 for Daymet, 2 for NLDAS, and 3 for Maurer), while the "+" connects the model*





*types used within an ensemble. The x-axis group and subscript "seed" indicate that*
*simulation results were averaged based on three different random seeds (see Figure C1).*
*Other points without "seed", along with their corresponding error bars, are derived from the*
*averages of metrics computed over repeated runs with three different random seeds. The*
*error bar indicates one standard deviation above and below the average value for each*
*simulation.*

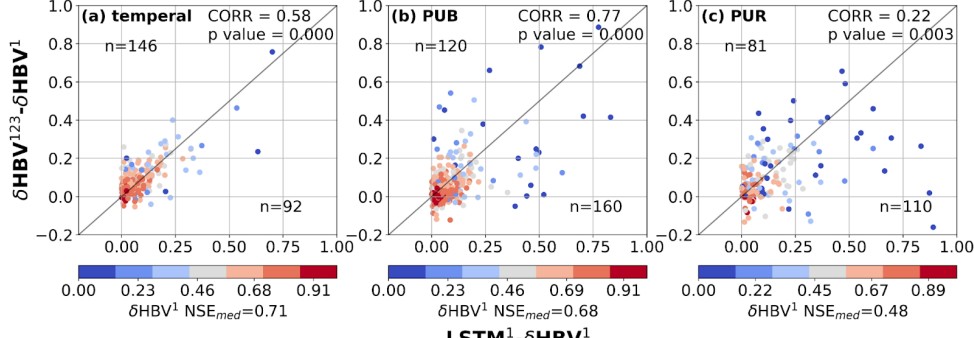


*Figure 3. Scatter plots comparing the performance differences between hydrological models*
*for the basins where LSTM outperformed δHBV (the basins where δHBV outperformed are*
*not shown in this plot). The x-axis represents the NSE differences between $LSTM^1$ and*
*$δHBV^1$ ($LSTM^1$ - $δHBV^1$), while the y-axis shows the NSE differences between $δHBV^{123}$ and*
*$δHBV^1$ ($δHBV^{123}$- $δHBV^1$). Points are color-coded according to the NSE values of $δHBV^1$.*
*The correlation coefficient (CORR) and p values between x-axis values and y-axis values,*
*along with the median NSE value of $δHBV^1$ ($NSE_{med}$) on these basins are also noted. We*
*note that NSE is not additive and should in general not be subtracted. Here the purpose is*
*only to confirm that basins where LSTM outperforms δHBV also tend to be those that benefit*
*from the ensemble of forcings.*

In the lower-performing basins where LSTM[1] had advantages over δHBV[1], the ensemble
of meteorological forcings δHBV[123] also tended to be higher than δHBV[1] (Figure 3),
suggesting that forcing quality was a significant reason behind the underperformance of
δHBV[1] in these basins. Similar patterns were also observed when analyzing RMSE values
(Figure D1). These basins previously contributed to LSTM's cumulative distribution function



of NSE diverging from that of δHBV[1] at the low end (Feng et al., 2022). Forcing errors can
exist in the form of systematic timing errors, low or high bias for larger events, etc., which
can be difficult for the mass-balanced conceptual HBV[1] structure to adapt to these errors.
Because the ensemble of forcings tends to suppress the errors in each forcing source, part of
the advantages of δHBV[123] over δHBV[1] can be attributed to reducing forcing bias or timing
errors. Since the advantages of LSTM[1] over δHBV[1] also tend to occur with these same
basins, this also explains how LSTM[1] surpasses δHBV[1] in some basins with poorer-quality
forcings. In contrast to δHBV, LSTM has the innate ability to shift information in time and
moderately adjust the input scale. Moving from temporal validation to PUB to PUR
scenarios, the advantages of diverse forcing datasets appear to diminish, as evidenced by the
decreasing ratio of points above versus below the diagonal line, since the forcing error
patterns remembered by LSTM may not generalize well in space (discussed in more detail in
sect. 3.2).

Ensembling streamflow simulations from different meteorological forcing datasets
demonstrates certain advantages over the previous approach of simultaneously sending
multiple forcings into an ML model like LSTM (Kratzert et al., 2021). Ensembling LSTM
simulations each using a single forcing dataset ($LSTM^{123}$) resulted in an NSE value of
0.8082, higher than that of 0.7974 from feeding multiple forcing datasets into a single LSTM
($LSTM^{multi}$). This difference was more pronounced in the cross-model-type ensemble, after
including δHBV, compared to the previous within-class ensemble, and particularly notable
for the spatial generalization tests (to be discussed in more detail in Sect. 3.2), with specific
metric values provided in Tables D1-D5. These results indicate that the trained LSTM in
$LSTM^{multi}$ may be overfitted to the significant redundant information in these three forcing
datasets, and that only LSTM cannot fully exploit the information hidden in the multiple





forcing datasets. Training separate ensemble members via different nonlinear hydrological
processes, on the other hand, seems to allow different bias features to emerge with separate
forcing datasets, accordingly mitigating them during the subsequent ensembling process.

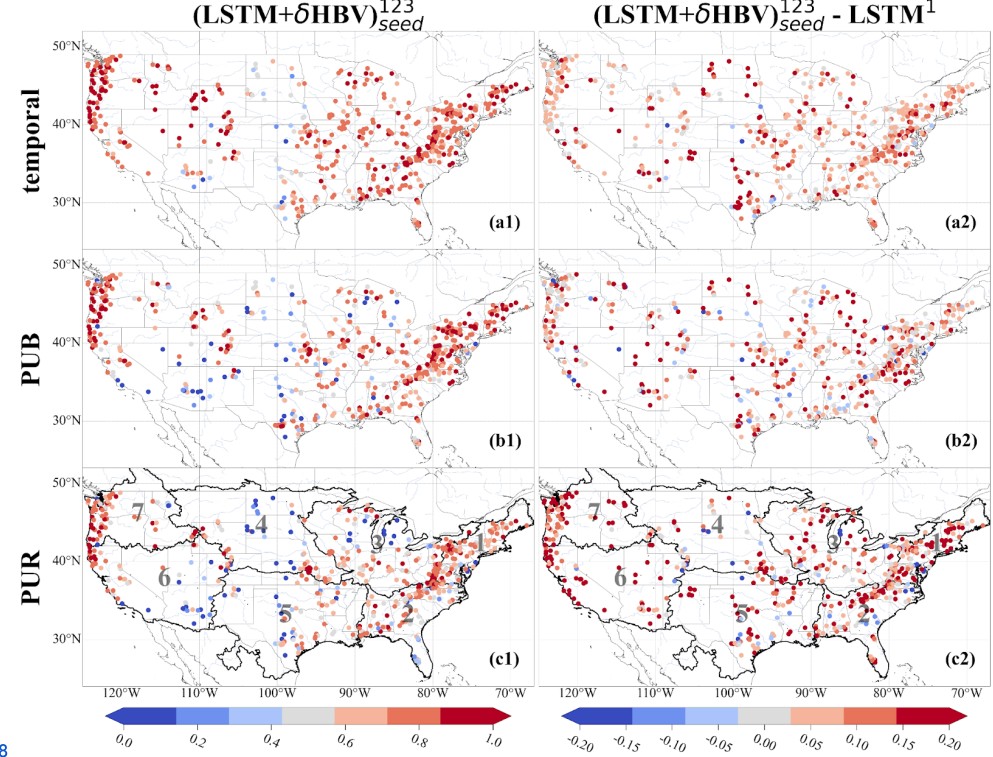


*Figure 4. Spatial distributions of NSE values over 531 basins. Subplots are arranged in rows,*
*indicating (a) temporal, (b) PUB, and (c) PUR test results, and columns, denoting (1) NSE*
*values from* $(LSTM + \delta HBV)_{seed}^{123}$ *and (2) the differences between these NSE values and*
*those of* $LSTM^1$ *(models using only forcing 1, Daymet). For* $LSTM^1$*, each NSE value reported*
*was the average of three NSE values from three simulations using three different random*
*seeds. The seven continuous regions used to divide up basins for the PUR test are outlined*
*and numbered in the PUR test maps.*

Our most diverse ensemble, $(LSTM + \delta HBV)_{seed}^{123} + LSTM_{seed}^{multi}$, achieved a median

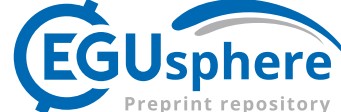

NSE value of ~0.83, surpassing the ~0.82 benchmark set by $LSTM_{seed}^{multi}$ (Table D4). This
advancement was achieved through random seed variation and cross-model-type ensembling.
The performance of $(LSTM + \delta HBV)^{123}$ ensemble proved more robust than $LSTM^{multi}$,
with only a slight boost when we incorporated random seeds, i.e., $(LSTM + \delta HBV)_{seed}^{123}$.
Notably, the derived $(LSTM + \delta HBV)_{seed}^{123}$ ensemble outperformed $LSTM^1$ across almost all
basins (Figure 4). Further incorporation of $LSTM^{multi}$ into this framework, especially when
using multiple random seeds, $(LSTM + \delta HBV)_{seed}^{123} + LSTM_{seed}^{multi}$, yielded the best overall
performance. Here, the margin over the previous benchmark was small in the temporal test.
However, as we will show in sect. 3.2, the previous benchmark, $LSTM_{seed}^{multi}$, lacked
robustness, exhibited greater deficiencies in spatial generalization, and negatively impacted
ensemble simulations.
When we changed the number of random seeds from 3 to 10, we found that although all
model and ensemble performances slightly increased, the gaps between them did not change
much (Figure 5 and Table D5). In particular, the gap between
$(LSTM + \delta HBV)_{seed}^{123} + LSTM_{seed}^{multi}$ and $(LSTM + \delta HBV)_{seed}^{123}$ or $LSTM_{seed}^{multi}$ remained
unchanged. This indicates that benefits from more random seeds rapidly become marginal,
and our results based on 3 random seeds were sufficiently robust. It was noteworthy that
while the $(LSTM + \delta HBV)^{123}$ ensemble generally showed the lowest RMSE values, it did
not always show the best high flow performance, as indicated by highRMSE (Tables D1-D4).
After incorporating the $LSTM_{seed}^{multi}$ variant into $(LSTM + \delta HBV)_{seed}^{123} + LSTM_{seed}^{multi}$, overall
RMSE and highRMSE both improved. Nevertheless, this ensemble did not always obtain the
best values in other metrics like low flow (lowRMSE) and requires further improvement.




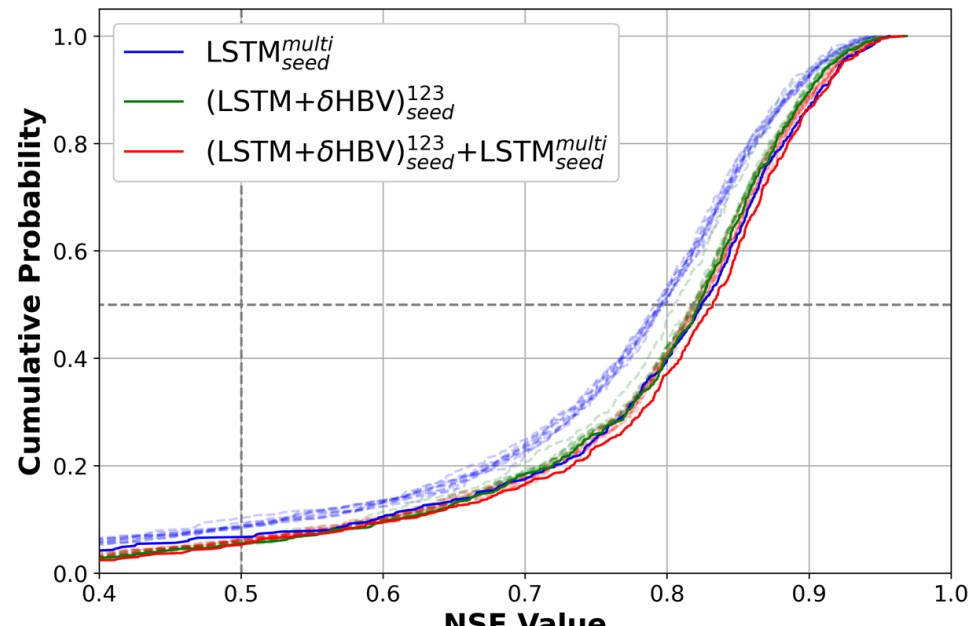


*Figure 5. Cumulative distribution function (CDF) curves based on temporal test results for $LSTM^{multi}$, $(LSTM + \delta HBV)^{123}$, and $[(LSTM + \delta HBV)^{123} + LSTM^{multi}]$. The solid lines (with "seed") denote the results with 10 random seeds while the corresponding dashed and translucent lines denote the performances of their individual members each based on one random seed.*

## 3.2. Spatial generalization

It is clear that cross-model-type ensembling and the incorporation of δHBV significantly improved prediction in ungauged basins (PUB) or regions (PUR), mitigating the difficulty of spatial generalization (Figure 2b - 2c). In particular, the previous record-holder for temporal test performance, $LSTM^{multi}_{seed}$, incurred large drops in the PUB and PUR tests, once again reminding us of the limitations of LSTM in spatial generalization. Given the same forcings, δHBV-only individual simulations or ensembles consistently outperformed LSTM-only counterparts in the PUR test. Furthermore, adding δHBV to the same-model-type LSTM



ensembles improved median NSE by 0.02-0.03 for PUB. The role of δHBV became even
more prominent in the harder PUR tests, with an increased gap (0.04-0.07), e.g., LSTM[123]
(median NSE ~0.656) and $(LSTM + δHBV)^{123}$ (median NSE ~0.701). The increased
significance of δHBV is also illustrated by the optimized weights shown in Figure E1. The
weights are estimated via a genetic algorithm using the streamflow observations during test
periods. Here the estimated weights are solely used to display the relative contributions of
different ensemble components. The significantly different spatial distribution patterns of
these weights among different test scenarios also indicate the differences among temporal,
PUB, and PUR tests (Figures E2-E3). The performance of $(LSTM + δHBV)^{123}$ improved
compared to $LSTM^{multi}$ whether or not we employed multiple random seeds to form an
ensemble. As such, we can conclude that the inclusion of a differentiable process-based
model like δHBV in an ensemble is a systematic way to reduce the risks of failed
generalizations of LSTM.
Utilizing a cross-model-type ensemble led to widespread improvements over
LSTM-only ensembles, with the exception of a few scattered basins for each temporal
(Figure 4-a2), PUB (Figure 4-b2), and PUR (Figure 4-c2) test. The most significant
improvements due to the ensemble were concentrated on the center of the Great Plains along
with the midwestern US, while the eastern US was moderately improved, suggesting data
uncertainty is a larger issue in the central and midwestern US. The Great Plains have
historically had poor performance for all kinds of models (Mai et al., 2022) and even the
ensemble model had NSE values of only 0.3-0.4 for many of the basins there, although this
still marked significant improvements over LSTM[1] (Figure 4-a2, -b2, -c2). Some western
basin NSE values were elevated by more than 0.15 for the temporal test (Figure 4-a2) and
even more for PUB and PUR. Meteorological stations are generally sparse on the Great
Plains and an ensemble seems to be an effective way to leverage the different forcing datasets



that are available. The poor performances in some basins highlight some remaining
deficiencies in current models which clearly cannot fully consider the heterogeneities of
different basins; thus, multiscale formulations that resolve such heterogeneities may have
advantages (Song et al., 2024a).
To investigate why ensembles outperformed single-model, single-forcing approaches,
we compared their temporal, PUB, and PUR test simulation time series against observations
for 531 basins (Figure 6). Analysis of averaged hydrological year data revealed that while
individual ensemble members using single-source forcing datasets performed similarly for
easily simulated periods, they showed significant divergence during challenging periods,
particularly peak flows. This divergence stems from distinct systematic errors inherent to
different model types and forcing datasets. Notably, LSTM-based simulations alone proved
insufficient in generating adequate spread to capture these divergent points. A key finding
was that δHBV exhibited markedly different variation patterns compared to LSTM, and its
inclusion substantially increased the ensemble spread. By averaging individual model outputs
and stabilizing uncertainties, ensemble simulations achieved effective and robust
performance across all conditions, which can be shown via the metric highRMSE and
lowRMSE values in Tables D1-D4. This highlights the critical importance of comprehensive
training for each ensemble member to enable the development of distinct characteristics in
their streamflow simulations, ultimately enhancing ensemble performance.



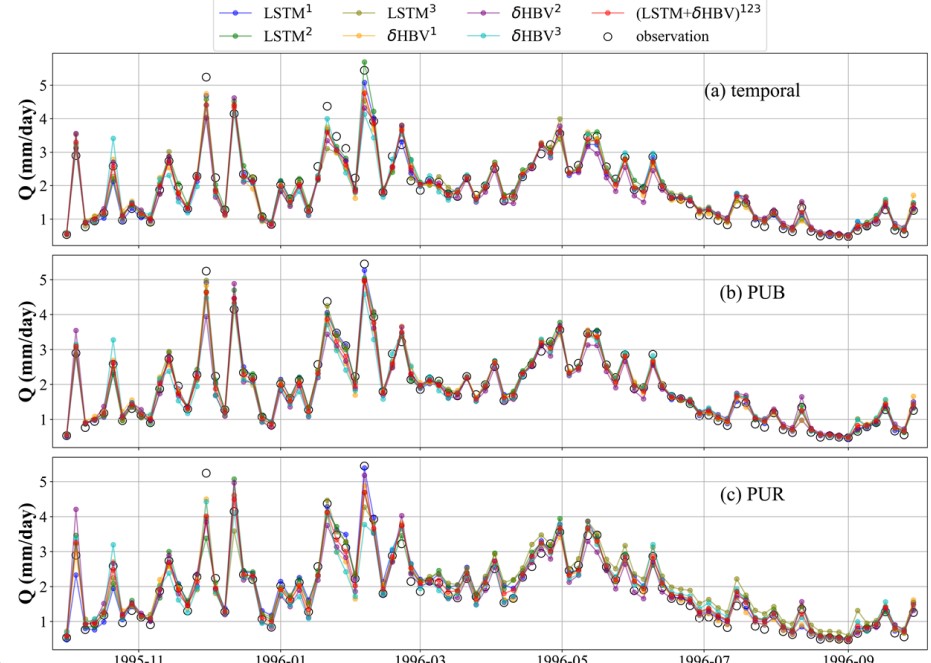

*Figure 6. Comparisons between multi-basin-averaged streamflow observations and simulations across 531 basins. The time series points are displayed at four-day intervals for clarity and conciseness.*

3.3. Further discussion

Based on our results, we identified several avenues for future research. First, while we have explored various weighting strategies and found that averaging yields the best performance yet, we believe that dynamic or adaptive weighting schemes could further enhance performance in future studies. It is also demonstrated by Table E1 that estimated uneven weights can significantly improve simulation performance. Moreover, within specific basins, the estimated weights of different components are often highly imbalanced, as evidenced by the spatial distribution of optimized weights (Figures E2-E3). Some potential feasible ways include using the simulations from these individual trained models as inputs of a data-driven model (Solanki et al., 2025), and making the weight estimation and the ensemble member training simultaneously.



Both LSTM and δHBV models exhibit limitations in regions with significant
anthropogenic impacts like dam presence, as well as arid climatic and significantly
heterogeneous geological conditions. Further improvements may include incorporating
additional data that capture these factors like capacity-to-runoff ratio (Ouyang et al., 2021) or
integrating specialized modules, such as reservoirs (Hanazaki et al., 2022; West et al., 2025).
Compared with LSTM, δHBV is more sensitive to precipitation biases. For example, the
differences between δHBV simulations under different forcing datasets were generally larger
than those for LSTM, and δHBV using the Daymet forcing dataset showed largely better
performance than with the other two forcing datasets, which indicates that δHBV may not be
able to fit different forcing datasets well. Therefore, many potential structural optimizations
can be implemented to improve δHBV. Our analysis provided corroborating evidence that
forcing error is an important reason why LSTM can outperform δHBV in the temporal test
for some basins, although such patterns may not generalize well in space. A meteorological
forcing data correction module can be developed in the future to account for timing and
magnitude errors in precipitation. Moreover, ensemble simulations may face challenges when
computational resources are limited and calculations are performed sequentially. However,
we remain optimistic about these challenges, as the processes can be addressed by leveraging
parallel computing with multiple GPUs, benefiting from ongoing advancements in
computational power.
For this work, we did not create a δHBV$^{multi}$ model (in the same vein as LSTM$^{multi}$) using
all forcings as an input to a single model, since a similar experiment has already been
conducted by Sawadekar et al. (2024). We also did not examine "seed" combinations of a
δHBV$^{multi}$ as we believed they would not result in a significant performance boost (unlike that
seen with LSTM$^{multi}$), because LSTM has high variability and low bias, while δHBV has
lower variance and potentially higher bias. As a result, random seeds would likely not create



large enough perturbations for δHBV and wouldn't bring the benefits seen with $LSTM_{seed}^{multi}$.
To achieve an equivalent perturbation level for δHBV, it may be necessary to incorporate
multiple distinct hydrological models, such as SAC-SMA, PRMS, and GR4J, similar to the
approach implemented in the Framework for Understanding Structural Errors (FUSE) (Clark
et al., 2008). Work is ongoing to create a combination of a series of differentiable
process-based models, which is expected to produce a further improved ensemble with great
interpretability. Given the success of cross-model-type ensembles shown in this work, we
also encourage further exploration of ensemble simulations involving models with other
distinct mechanisms.

## 521 4. Summary and Conclusions

This study comprehensively analyzes ensemble combinations of two advanced model
types (LSTM and δHBV), each with distinct mechanisms, for streamflow simulation across
basins in the US. Three meteorological forcing datasets (Daymet, NLDAS, and Maurer)
are employed to fully capture the characteristics of the two models, and their applications in
two different ways are also tested. The performance of ensemble simulations was evaluated
under three distinct testing scenarios (temporal, PUB, and PUR tests), surpassing the previous
highest performances. Our findings enhance the understanding of how to effectively utilize
diverse model types and multi-source datasets to improve streamflow simulations. The
principal conclusions are:
(1) Cross-model-type    ensembles    (LSTM+δHBV)    consistently    outperformed

single-model approaches across all test scenarios, setting new performance

benchmarks    on    the    CAMELS    dataset.    These    ensembles    demonstrated    the

complementarity of data-driven (LSTM) and physics-informed (δHBV) approaches

in capturing diverse hydrological behaviors.



(2) Ensembling models trained on different forcing datasets proved more effective than using multiple forcing datasets as simultaneous inputs to a single model. This suggests that separate training allows each model to capture unique features contained in each forcing dataset, which can then be effectively leveraged in the ensemble.

(3) δHBV provided significant benefits to ensemble simulations on spatial generalization. Ensembling LSTM with δHBV showed increasing benefits as generalization challenges increased, from temporal to spatial interpolation (PUB) to spatial extrapolation (PUR) tests. This underscores the value of physics-informed constraints in improving model transferability to ungauged basins and regions.

(4) While ensemble methods significantly improved overall performance, they did not fully mitigate consistent deficiencies in certain challenging areas (e.g., regions with high dam density or heterogeneous hydrogeological conditions). This indicates areas for future model development.

These findings have important implications for hydrological modeling and water resources management. The improved accuracy and spatial generalization of our ensemble approach can enhance streamflow predictions, benefiting water resources planning and management, particularly in data-scarce regions. Our results also suggest that future hydrological model development should focus on combining data-driven and physics-based approaches to improve model generalizability across diverse conditions. The superior performance of ensembling models with different forcing datasets over using merged forcings as a single input highlights the risk of indiscriminately feeding all available data into one data-driven model. While computational demands certainly require consideration, the potential improvements in prediction accuracy offer significant value for both research and operational applications. Future work should focus on refining these ensemble techniques,



addressing model limitations in challenging regions, and exploring ensemble implementation
in operational settings.



**Code and data availability**
The source codes and datasets utilized in this study are publicly accessible through the
following repositories: The δHBV modeling framework, including all computational scripts
and documentation, is hosted on Zenodo (https://doi.org/10.5281/zenodo.7091334) (Feng et
al., 2023a), with an updated version and comprehensive software release scheduled upon
manuscript acceptance. The implementation of the LSTM architecture is accessible through
Zenodo (https://doi.org/10.5281/zenodo.6326394) (Kratzert et al., 2022). The CAMELS
hydrometeorological dataset, which provides the foundational basin characteristics and time
series data used in our analysis, can be obtained via https://dx.doi.org/10.5065/D6MW2F4D
(Addor et al., 2017; Newman and Clark, 2014).

**Author contributions**
PL and CS designed the experiments and PL carried them out. YS developed the
modified δHBV code. PL prepared the manuscript with contributions from all co-authors.

**Competing interests**
Chaopeng Shen and Kathryn Lawson have financial interests in HydroSapient, Inc., a
company that could potentially benefit from the results of this research. This interest has been
reviewed by the Pennsylvania State University in accordance with its individual conflict of
interest policy for the purpose of maintaining the objectivity and the integrity of research.



The other authors have no competing interests to declare.

## Acknowledgments

PL, CS, and KL were supported by the Office of Biological and Environmental Research of the U.S. Department of Energy (contract no. DESC0016605). PJ and MP were also partially supported by California Department of Water Resources Atmospheric River Program Phase III (Grant 4600014294). YS and CS were partially supported by subaward A23-0252-S002 from the Cooperative Institute for Research to Operations in Hydrology (CIROH) through the National Oceanic and Atmospheric Administration (NOAA) Cooperative Agreement (Grant no. NA22NWS4320003).

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



**Appendix A: Detailed processes of HBV employed in this study.**

The Hydrologiska Byrans Vattenbalansavdelning (HBV) model (Aghakouchak and Habib, 2010; Beck et al., 2020; Bergström, 1976, 1992) is a simple but effective bucket-type hydrologic model that simulates hydrologic variables including snow water equivalent, soil water, groundwater storage, evapotranspiration, quick flow, baseflow, and total streamflow. In the following texts, we describe these processes in detail by equations, in which uppercase letters indicate state variables, and lowercase letters indicate model parameters. In general, the water balance is developed based on Equation (S1).

$$EP - AE - Q_t = SN + SM + UR + LR + LAKE \tag{S1}$$

where $EP$ is effective precipitation, $AE$ is the actual evapotranspiration, $Q_t$ is the total simulated runoff, $SN$ is snow, $SM$ is soil water storage, $UR$ is the upper reservoir water level, $LR$ is the lower reservoir water level, and $LAKE$ is the lake level (omitted in this study). First, $EP$ is separated into liquid ($RN$) and solid ($SN$) components based on the temperature ($T$) relative to the threshold temperature ($tt$) as

$$RN = EP \ if \ T \geq tt \tag{S2}$$

$$SN = EP \ if \ T < tt \tag{S3}$$

Snow ($SN$) accumulates in the snowpack ($SNP$), while the snowmelt ($SNM$) is calculated using a temperature-dependent melt rate ($cfm$). The snowmelt ($SNM$) is limited to the available snowpack ($SNP$), and any excess melt contributes to meltwater ($MW$) as

$$SNP = SNP + SN \tag{S4}$$

$$SNM = \begin{cases} SNP & cfm \cdot (T - tt) \geq SNP \\ cfm \cdot (T - tt) & T \geq tt, cfm \cdot (T - tt) < SNP \\ 0 & T < tt \end{cases} \tag{S5}$$

$$MW = MW + SNM \tag{S6}$$

$$SNP = SNP - SNM \tag{S7}$$

Some of this meltwater ($MW$) refreezes based on a refreezing parameter ($cfr$) and the temperature difference from the threshold, returning to the snowpack ($SNP$). The amount of refrozen water is labeled as $FRZ$.

$$RFZ = \begin{cases} MW & cfr \cdot cfm \cdot (tt - T) \geq MW \\ cfr \cdot cfm \cdot (tt - T) & T < tt, cfr \cdot cfm \cdot (tt - T) < MW \\ 0 & T \geq tt \end{cases} \tag{S8}$$



$$SNP = SNP + RFZ \tag{S9}$$

$$MW = MW - RFZ \tag{S10}$$

The remaining meltwater ($MW$) that exceeds the snowpack's holding capacity ($cwh$)
contributes to soil infiltration ($IF$), and the rest remains in the meltwater ($MW$) storage as

$$IF = \begin{cases} MW - cwh * SNP & MW - cwh * SNP \geq 0 \\ 0 & MW - cwh * SNP < 0 \end{cases} \tag{S11}$$

$$MW = MW - IF \tag{S12}$$

The fraction of soil moisture relative to the field capacity ($fc$) determines the soil wetness,
which modulates the amount of water recharged into the soil ($SP$). Then soil moisture ($SM$) is
updated based on the infiltration of meltwater ($IF$), rain ($RN$), and the amount of recharged
water ($SP$) as

$$SP = \left(\frac{SM}{fc}\right)^{\beta} \cdot (IF + RN) \tag{S13}$$

$$SM = SM + IF + RN - SP \tag{S14}$$

The excess water, above the field capacity ($IF_{dir}$), is calculated and subsequently removed
from the soil moisture storage as

$$IF_{dir} = \begin{cases} SM - fc & if \cdot SM \geq fc \\ 0 & if \cdot SM < fc \end{cases} \tag{S15}$$

$$SM = SM - IF_{dir} \tag{S16}$$

Actual evapotranspiration ($AE$) is determined by an evaporation factor ($PEC$), which depends
on the soil moisture, a shape parameter ($\lambda$), a parameter ($lp$), and field capacity ($fc$) for
evapotranspiration. This factor limits the actual evapotranspiration ($AE$) to both the potential
evapotranspiration ($PE$) and the available soil moisture.

$$PEC = \begin{cases} \left(\frac{SM}{lp \cdot fc}\right)^{\lambda} & if\ 0 \leq \left(\frac{SM}{lp \cdot fc}\right)^{\lambda} < 1 \\ 0 & if\ S\left(\frac{SM}{lp \cdot fc}\right)^{\lambda} < 0 \\ 1 & if\ S\left(\frac{SM}{lp \cdot fc}\right)^{\lambda} \geq 1 \end{cases} \tag{S17}$$

$$AE = \begin{cases} PE \cdot PEC & if\ SM \geq PE \cdot PEC \\ SM & if\ SM < PE \cdot PEC \end{cases} \tag{S18}$$





$$SM = SM - AE \tag{S19}$$

Capillary rise ($CP$) from the lower soil zone ($SLZ$) is governed by a parameter ($c$), which
determines the amount of water moving upward based on the soil moisture content. This
capillary flow replenishes the soil moisture, while groundwater interactions occur through
recharge processes in the upper ($SUZ$) and lower ($SLZ$) groundwater zones.

$$CP = \begin{cases} SLZ & if \cdot SLZ < c \cdot SLZ \cdot (1 - \dfrac{SM}{fc}) \\ c \cdot SLZ \cdot (1 - \dfrac{SM}{fc}) & if \cdot SLZ \geq c \cdot SLZ \cdot (1 - \dfrac{SM}{fc}) \end{cases} \tag{S20}$$

$$SM = SM + CP \tag{S21}$$

$$SLZ = \begin{cases} SLZ - CP & if \cdot SLZ \geq CP \\ 0 & if \cdot SLZ < CP \end{cases} \tag{S22}$$

Excess recharge ($SP$ and $IF_{dir}$) from the soil enters the upper zone, where it either percolates
to the lower zone ($PERC$) based on a constant rate ($prc$) or contributes to direct runoff ($Q_0$)
when it exceeds the upper zone threshold ($uzl$). The generated flow is modeled using
parameters ($k_0$, $k_1$, $k_2$) governing flow from the upper and lower zones. Each of these flows
contributes to runoff ($Q_0$, $Q_1$, $Q_2$), and their respective contributions to streamflow ($Q_t$) are
modeled over time.

$$SUZ = SUZ + SP + IF_{dir} \tag{S23}$$

$$PERC = \begin{cases} SUZ & if \cdot SUZ \geq prc \\ prc & if \cdot SUZ < prc \end{cases} \tag{S24}$$

$$SUZ = SUZ - PERC \tag{S25}$$

$$Q_0 = \begin{cases} k_0 \cdot (SUZ - uzl) & if \cdot SUZ \geq uzl \\ 0 & if \cdot SUZ < uzl \end{cases} \tag{S26}$$

$$SUZ = SUZ - Q_0 \tag{S27}$$

$$Q_1 = SUZ \cdot k_1 \tag{S28}$$

$$SUZ = SUZ - Q_1 \tag{S29}$$

$$SLZ = SLZ + PERC \tag{S30}$$



$$Q_2 = SLZ \cdot k_2 \tag{S31}$$

$$SLZ = SLZ - Q_2 \tag{S32}$$

$$Q_t = Q_0 + Q_1 + Q_2 \tag{S33}$$


Finally, a routing module (Feng et al., 2022) is used to process $Q_t$ to produce the final
streamflow output ($Q_t^*$). This module with two parameters ($\theta_\alpha$, $\theta_\tau$) assumes a gamma function
for the unit hydrograph and convolves the unit hydrograph with the runoff as,

$$Q_t^* = \int_0^{tmax} \xi(s: \theta_\alpha, \theta_\tau) \cdot Q(t - s)\,ds \tag{S34}$$

$$\xi(s: \theta_\alpha, \theta_\tau) = \frac{1}{\Gamma(\theta_\alpha)\theta_\tau^{\theta_\alpha}} t^{\theta_\alpha - 1} e^{-\frac{t}{\theta_\tau}} \tag{S35}$$







**Appendix B: Illustrated differences among the three meteorological forcing datasets**

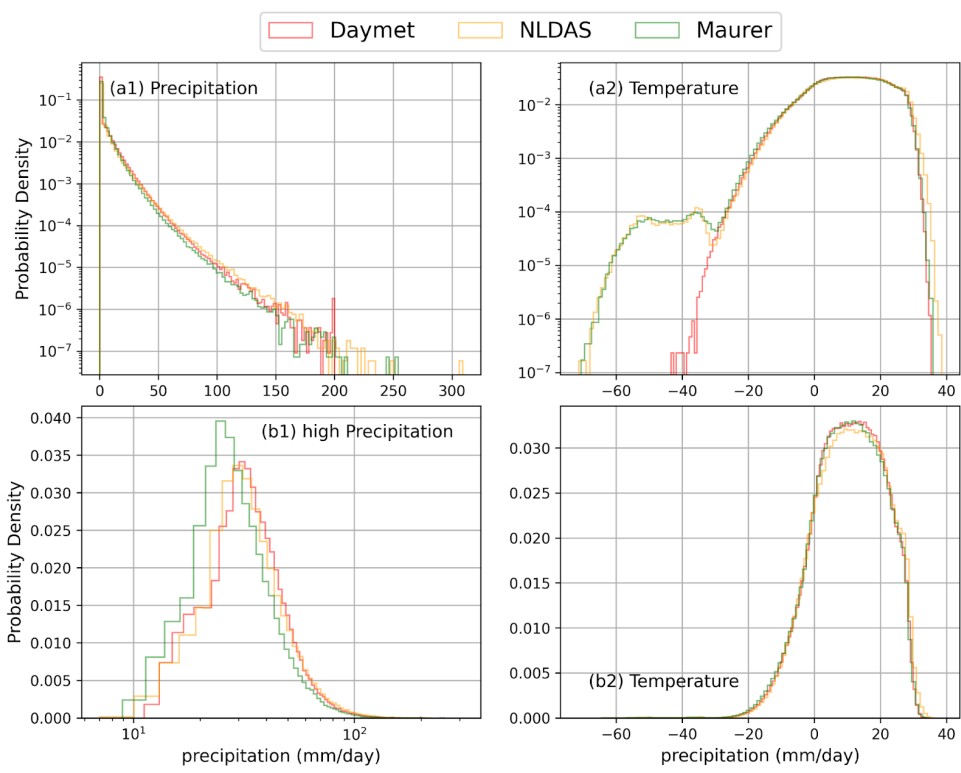

*Figure B1. Probability density distributions of precipitation and temperature across three meteorological forcing datasets.*



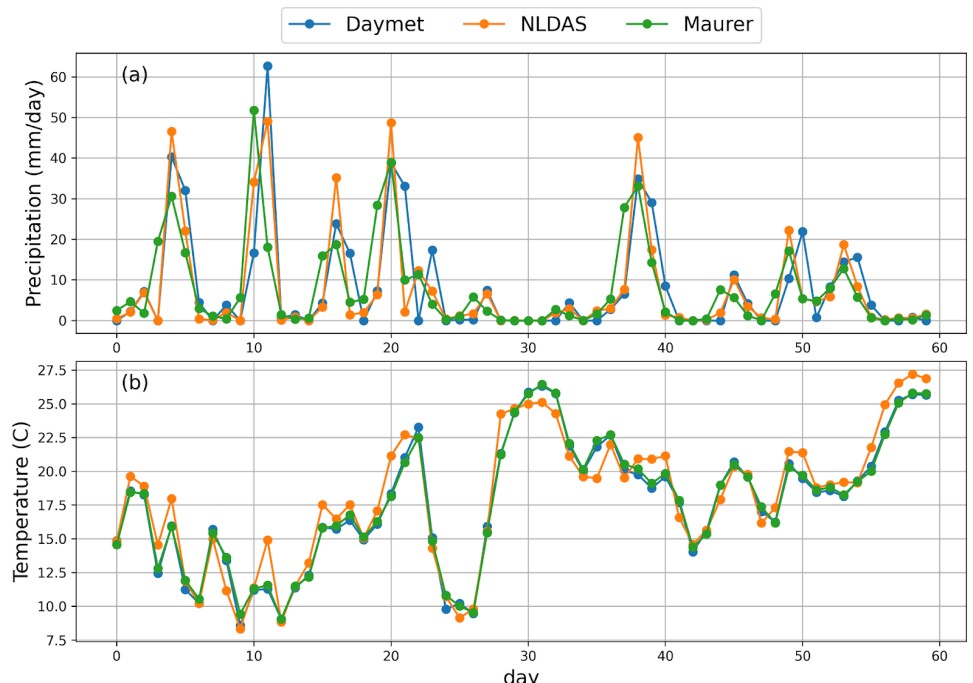

Figure B2. *Illustrated temporal variations of precipitation and temperature in a basin across three meteorological forcing datasets.*



**Appendix C: Details of model inputs, ensemble frameworks, and evaluations**

*Table C1. Full names for the abbreviations of dynamic data (all but streamflow are "forcings") and static basin attributes used as model inputs. All variables and their values are provided in the CAMELS dataset (Addor et al., 2017) except for the NLDAS and Maurer daily temperature extrema, which are from Kratzert et al. (2021). Potential evapotranspiration and normalized streamflow were calculated for the purposes of this work, using CAMELS data.*

| Type | Abbreviation | Full name | Unit |
|---|---|---|---|
| **Dynamic data** | prcp | Precipitation | mm/day |
| | pet | Potential evapotranspiration (calculated in this work using the Hargreaves equation and CAMELS data) | mm/day |
| | tmean | Mean air temperature | °C |
| | tmax | Maximum air temperature | °C |
| | tmin | Minimum air temperature | °C |
| | dayl | Day length | s/day |
| | srad | Shortwave radiation | W/m$^2$ |
| | vp | Water vapor pressure | pa |
| | q_vol | Volumetric streamflow | ft3/s |
| | q | Streamflow normalized by basin area (q_vol / area_gages2) | mm/day |
| **Static basin attributes** | p_mean | Mean daily precipitation | mm/day |
| | pet_mean | Mean daily potential evapotranspiration | mm/day |
| | p_seasonality | Seasonality and timing of precipitation | - |
| | frac_snow | Fraction of precipitation falling as snow | - |
| | aridity | Rate of mean values of potential evapotranspiration precipitation | - |
| | high_prec_freq | Frequency of high precipitation days | days/year |
| | high_prec_dur | Average duration of high precipitation events | days |
| | low_prec_freq | Frequency of dry days | days/year |
| | low_prec_dur | Average duration of dry periods | days |



| | | | |
|---|---|---|---|
| | elev_mean | Catchment mean elevation | m |
| | slope_mean | Catchment mean slope | m/km |
| | area_gages2 | Catchment area (GAGES-II estimate) | km$^2$ |
| | frac_forest | Fraction of catchment area having land cover identified as forest | - |
| | lai_max | Maximum monthly mean of the leaf area index | - |
| | lai_diff | Difference between the maximum and minimum monthly mean of the leaf area index | - |
| | gvf_max | Maximum monthly mean of the green vegetation | - |
| | gvf_diff | Difference between the maximum and minimum monthly mean of the green vegetation fraction | - |
| | dom_land_cover_frac | Fraction of the catchment area associated with the dominant land cover | - |
| | dom_land_cover | Dominant land cover type | - |
| | root_depth_50 | Root depth at 50$^{th}$ percentile, extracted from a root depth distribution based on the International Geosphere-Biosphere Programme (IGBP) land cover | m |
| | soil_depth_pelletier | Depth to bedrock | m |
| | soil_depth_statsgso | Soil depth | m |
| | soil_porosity | Volumetric soil porosity | - |
| | soil_conductivity | Saturated hydraulic conductivity | cm/hr |
| | max_water_content | Maximum water content | m |
| | sand_frac | Fraction of soil which is sand | - |
| | silt_frac | Fraction of soil which is silt | - |
| | clay_frac | Fraction of soil which is clay | - |
| | geol_class_1st | Most common geologic class in the catchment basin | - |



| | | | |
|---|---|---|---|
| | geol_class_1st_frac | Fraction of the catchment area associated with its most common geologic class | - |
| | geol_class_2nd | Second most common geologic class in the catchment basin | - |
| | geol_class_2nd_frac | Fraction of the catchment area associated with its 2nd most common geologic class | - |
| | carbonate_rocks_frac | Fraction of the catchment area as carbonate sedimentary rocks | - |
| | geol_porosity | Subsurface porosity | - |
| | geol_permeability | Subsurface permeability | $m^2$ |





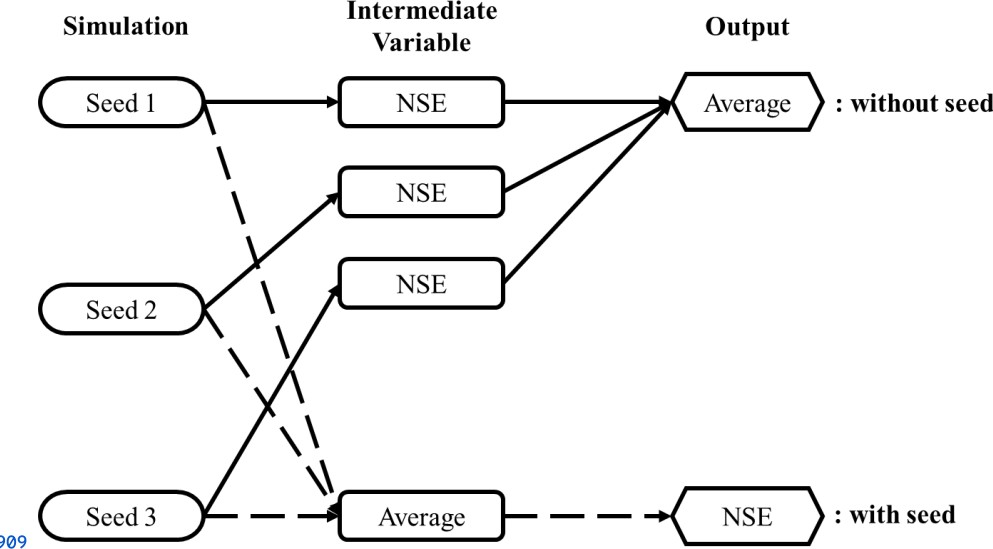


*Figure C1. Ensemble frameworks to generate metrics for ensembles named without (solid arrows) and with (dashed arrows) "seed" as a subscript.*






*Table C2. Evaluation metrics.*

| Statistic | Equation* |
|-----------|-----------|
| NSE | $$NSE = 1 - \frac{\sum_{i=1}^{n}(O_i - S_i)^2}{\sum_{i=1}^{n}(O_i - \mu_o)^2}$$ |
| KGE | $$KGE = 1 - \sqrt{(r-1)^2 + (\beta-1)^2 + (\gamma-1)^2},$$ $$\beta = \frac{\mu_S}{\mu_O}, \gamma = \frac{CV_S}{CV_O} = \frac{\sigma_S/\mu_S}{\sigma_O/\mu_O}$$ |
| PBIAS | $$\frac{\sum_{i=1}^{n}(O_i - S_i)}{\sum_{i=1}^{n}O_i} \times 100$$ |
| RMSE | $$\sqrt{\frac{1}{n}\sum_{i=1}^{n}(O_i - S_i)^2}$$ |

* *S is a streamflow simulation; O is the corresponding observation; n is the number of total S*
*or O; r is the linear Pearson correlation between S and O; $\beta$ is the mean bias; and $\gamma$ is the*
*variability bias. The mean and standard deviation of simulations are denoted as $\mu_S$ and $\sigma_S$,*
*respectively, and $\mu_O$ and $\sigma_O$ are the mean and standard deviation of the observations.*



*Table C2 (continued). Evaluation metrics.*

| Statistic | Range | Optimal Value |
|-----------|-------|---------------|
| NSE | -∞ to 1.0 | 1.0 |
| KGE | -∞ to 1.0 | 1.0 |
| PBIAS | -∞ to ∞ | 0.0 |
| RMSE | 0.0 to ∞ | 0.0 |

*\* S is a streamflow simulation; O is the corresponding observation; n is the number of total S*
*or O; r is the linear Pearson correlation between S and O; $\beta$ is the mean bias; and $\gamma$ is the*
*variability bias. The mean and standard deviation of simulations are denoted as $\mu_S$ and $\sigma_S$,*
*respectively, and $\mu_O$ and $\sigma_O$ are the mean and standard deviation of the observations.*





**Appendix D: Additional details on model performance**

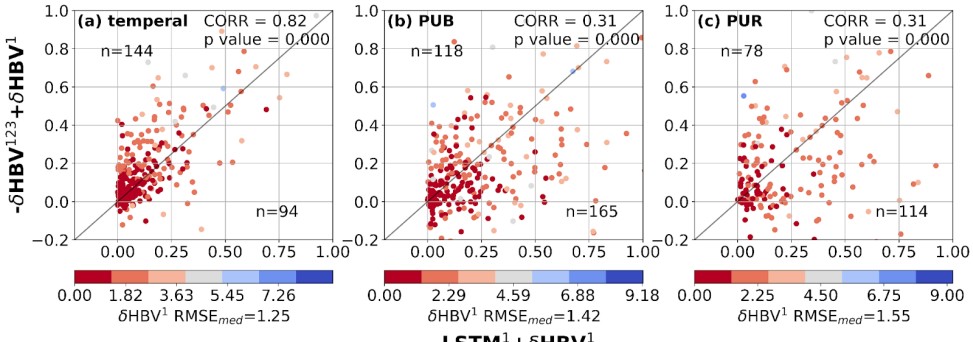


*Figure D1. Scatter plots comparing the performance differences between hydrological*
*models. The x-axis represents the RMSE differences between $LSTM^1$ and $\delta HBV^1$ ($LSTM^1$ -*
*$\delta HBV^1$), while the y-axis shows the RMSE differences between $\delta HBV^{123}$ and $\delta HBV^1$ (*
*$\delta HBV^{123}$-$\delta HBV^1$). Points are color-coded according to the RMSE values of $\delta HBV^1$. The*
*median NSE values of $\delta HBV^1$ ($RMSE_{med}$) on these basins are also noted.*






*Table D1. Median NSE, KGE, RMSE, PBIAS, and RMSE values under low (lowRMSE), high (highRMSE), and middle (midRMSE) flows based on 531 basins under the temporal test. The values are the mean of three simulations run with different random seeds.*

| Temporal | Number | Daymet | NLDAS | Maurer |
|---|---|---|---|---|
| LSTM | NSE | 0.735639 | 0.736301 | 0.717337 |
| | KGE | 0.789375 | 0.782555 | 0.760575 |
| | RMSE | 1.21088 | 1.19847 | 1.27723 |
| | PBIAS | 4.04818 | 5.99486 | 1.58911 |
| | lowRMSE | 0.0596913 | 0.0602381 | 0.0545577 |
| | highRMSE | 2.70508 | 2.89684 | 2.97028 |
| | midRMSE | 0.196039 | 0.210022 | 0.219922 |
| δHBV | NSE | 0.741641 | 0.719776 | 0.729142 |
| | KGE | 0.769522 | 0.733983 | 0.760453 |
| | RMSE | 1.17864 | 1.26864 | 1.22089 |
| | PBIAS | 4.65898 | -0.228925 | 3.14742 |
| | lowRMSE | 0.0598199 | 0.0646098 | 0.0627206 |
| | highRMSE | 2.6918 | 3.15195 | 2.71629 |
| | midRMSE | 0.228731 | 0.245014 | 0.230725 |
| LSTM+δHBV | NSE | 0.787871 | 0.793168 | 0.791637 |
| | KGE | 0.796322 | 0.783612 | 0.784754 |
| | RMSE | 1.07604 | 1.06746 | 1.06921 |
| | PBIAS | 4.82572 | 3.0815 | 3.19841 |



| | | | | |
|---|---|---|---|---|
| | lowRMSE | 0.0599687 | 0.0593688 | 0.0541188 |
| | highRMSE | 2.69665 | 2.82245 | 2.69425 |
| | midRMSE | 0.204261 | 0.218498 | 0.214325 |






*Table D1 (continued). Median NSE, KGE, RMSE, PBIAS, and RMSE values under low (lowRMSE), high (highRMSE), and middle (midRMSE) flows based on 531 basins under the temporal test. The values are the mean of three simulations run with different random seeds.*

| Temporal | Number | Daymet+NLDAS | Daymet+Maurer | NLDAS+Maurer | All |
|---|---|---|---|---|---|
| LSTM | NSE | 0.781275 | 0.791158 | 0.792144 | 0.808176 |
| | KGE | 0.800955 | 0.795026 | 0.794441 | 0.803476 |
| | RMSE | 1.09103 | 1.06374 | 1.06701 | 1.01395 |
| | PBIAS | 5.17159 | 3.34362 | 4.5305 | 4.48263 |
| | lowRMSE | 0.0636155 | 0.0582563 | 0.0566306 | 0.0613625 |
| | highRMSE | 2.70218 | 2.71366 | 2.78962 | 2.67803 |
| | midRMSE | 0.194849 | 0.199809 | 0.206653 | 0.197469 |
| δHBV | NSE | 0.786118 | 0.770939 | 0.777651 | 0.794455 |
| | KGE | 0.772697 | 0.776781 | 0.767756 | 0.776692 |
| | RMSE | 1.07984 | 1.12671 | 1.10878 | 1.05808 |
| | PBIAS | 1.85962 | 4.26278 | 1.79134 | 2.59063 |
| | lowRMSE | 0.0627661 | 0.0597778 | 0.0623739 | 0.0617863 |
| | highRMSE | 2.94274 | 2.73054 | 2.87583 | 2.84511 |
| | midRMSE | 0.231981 | 0.219738 | 0.228451 | 0.230136 |
| LSTM+δHBV | NSE | 0.8108 | 0.810476 | 0.812144 | 0.81866 |
| | KGE | 0.79586 | 0.796202 | 0.786088 | 0.794257 |
| | RMSE | 1.0162 | 1.01676 | 1.02515 | 1.00077 |
| | PBIAS | 4.13077 | 4.08096 | 3.26458 | 3.8972 |





|  | lowRMSE | 0.059935 | 0.0575384 | 0.0558506 | 0.0581869 |
|---|---|---|---|---|---|
|  | highRMSE | 2.76133 | 2.68642 | 2.78242 | 2.71392 |
|  | midRMSE | 0.208476 | 0.207761 | 0.213433 | 0.208582 |




*Table D2. Median NSE, KGE, RMSE, PBIAS, and RMSE values under low (lowRMSE), high*
*(highRMSE), and middle (midRMSE) flows based on 531 basins under the PUB test. The*
*values are the mean of three simulations run with different random seeds.*

| PUB | Number | Daymet | NLDAS | Maurer |
|---|---|---|---|---|
| LSTM | NSE | 0.702636 | 0.695496 | 0.694156 |
| | KGE | 0.693998 | 0.677438 | 0.6909 |
| | RMSE | 1.31714 | 1.3394 | 1.34233 |
| | PBIAS | 0.669018 | 0.283106 | 0.936582 |
| | lowRMSE | 0.087648 | 0.088393 | 0.086873 |
| | highRMSE | 4.2852 | 4.49292 | 4.16042 |
| | midRMSE | 0.354458 | 0.364921 | 0.368124 |
| δHBV | NSE | 0.706809 | 0.670636 | 0.682998 |
| | KGE | 0.703137 | 0.66566 | 0.686912 |
| | RMSE | 1.35541 | 1.41185 | 1.37942 |
| | PBIAS | 1.49234 | -2.43395 | 0.291966 |
| | lowRMSE | 0.0798196 | 0.0808967 | 0.0846775 |
| | highRMSE | 4.21648 | 4.49582 | 4.18003 |
| | midRMSE | 0.335159 | 0.351271 | 0.356903 |
| LSTM+δHBV | NSE | 0.74227 | 0.723778 | 0.72202 |
| | KGE | 0.715931 | 0.690154 | 0.707292 |
| | RMSE | 1.24887 | 1.278 | 1.26697 |
| | PBIAS | 1.27863 | -0.599778 | 0.903464 |



| | lowRMSE | 0.0816748 | 0.0795686 | 0.0825691 |
|---|---|---|---|---|
| | highRMSE | 4.08432 | 4.23483 | 3.94929 |
| | midRMSE | 0.327459 | 0.33851 | 0.347169 |






*Table D2 (continued). Median NSE, KGE, RMSE, PBIAS, and RMSE values under low (lowRMSE), high (highRMSE), and middle (midRMSE) flows based on 531 basins under the PUB test. The values are the mean of three simulations run with different random seeds.*

| PUB | Number | Daymet+NLDAS | Daymet+Maurer | NLDAS+Maurer | All |
|---|---|---|---|---|---|
| LSTM | NSE | 0.757853 | 0.749151 | 0.753136 | 0.768181 |
| | KGE | 0.713319 | 0.720099 | 0.716497 | 0.727143 |
| | RMSE | 1.18251 | 1.22254 | 1.19718 | 1.15026 |
| | PBIAS | 0.320396 | 0.931656 | 0.766216 | 0.970047 |
| | lowRMSE | 0.0875191 | 0.0864129 | 0.0835341 | 0.0874717 |
| | highRMSE | 4.1296 | 4.06602 | 4.17217 | 4.0061 |
| | midRMSE | 0.334683 | 0.349856 | 0.342819 | 0.333534 |
| δHBV | NSE | 0.748916 | 0.734052 | 0.733955 | 0.757749 |
| | KGE | 0.699768 | 0.714323 | 0.69436 | 0.714048 |
| | RMSE | 1.26852 | 1.27637 | 1.27244 | 1.23229 |
| | PBIAS | 0.0446112 | 1.212 | -1.04135 | 0.201809 |
| | lowRMSE | 0.0808293 | 0.0792486 | 0.0814476 | 0.0808359 |
| | highRMSE | 4.19575 | 3.97788 | 4.21623 | 4.07419 |
| | midRMSE | 0.311826 | 0.33668 | 0.339257 | 0.318165 |
| LSTM+δHBV | NSE | 0.780625 | 0.764866 | 0.767761 | 0.785833 |
| | KGE | 0.719781 | 0.725373 | 0.715982 | 0.723972 |
| | RMSE | 1.14924 | 1.17659 | 1.16881 | 1.13591 |
| | PBIAS | 0.186062 | 0.881644 | 0.405548 | 0.565489 |



| | lowRMSE | 0.0805946 | 0.0814251 | 0.0817114 | 0.0826379 |
|---|---|---|---|---|---|
| | highRMSE | 3.97373 | 3.86834 | 3.88 | 3.91692 |
| | midRMSE | 0.313708 | 0.324777 | 0.324089 | 0.323671 |







*Table D3. Median NSE, KGE, RMSE, PBIAS, and RMSE values under low (lowRMSE), high*
*(highRMSE), and middle (midRMSE) flows based on 531 basins under the PUR test. The*
*values are the mean of three simulations run with different random seeds.*

| PUR | Number | Daymet | NLDAS | Maurer |
|---|---|---|---|---|
| LSTM | NSE | 0.578365 | 0.546217 | 0.56164 |
| | KGE | 0.557788 | 0.559986 | 0.567231 |
| | RMSE | 1.59111 | 1.63626 | 1.5833 |
| | PBIAS | -0.575328 | -2.77709 | -0.623183 |
| | lowRMSE | 0.124837 | 0.118971 | 0.118695 |
| | highRMSE | 5.42346 | 5.38886 | 5.05212 |
| | midRMSE | 0.498133 | 0.498442 | 0.471744 |
| δHBV | NSE | 0.622278 | 0.592306 | 0.59161 |
| | KGE | 0.638818 | 0.601338 | 0.620877 |
| | RMSE | 1.57189 | 1.61191 | 1.63628 |
| | PBIAS | 1.27223 | -1.60075 | 1.62709 |
| | lowRMSE | 0.10142 | 0.102975 | 0.101075 |
| | highRMSE | 5.07706 | 5.16093 | 4.99602 |
| | midRMSE | 0.447879 | 0.474516 | 0.439697 |
| LSTM+δHBV | NSE | 0.644398 | 0.618255 | 0.635444 |
| | KGE | 0.627481 | 0.605237 | 0.615883 |
| | RMSE | 1.46185 | 1.5153 | 1.48393 |
| | PBIAS | -0.269697 | -0.719505 | 0.197859 |





|  | lowRMSE | 0.105146 | 0.100944 | 0.106272 |
|---|---|---|---|---|
|  | highRMSE | 4.95749 | 4.99478 | 4.78638 |
|  | midRMSE | 0.431456 | 0.4575 | 0.426126 |





*Table D3 (continued). Median NSE, KGE, RMSE, PBIAS, and RMSE values under low (lowRMSE), high (highRMSE), and middle (midRMSE) flows based on 531 basins under the PUR test. The values are the mean of three simulations run with different random seeds.*

| PUR | Number | Daymet+NLDAS | Daymet+Maurer | NLDAS+Maurer | All |
|---|---|---|---|---|---|
| LSTM | NSE | 0.634398 | 0.636369 | 0.626939 | 0.656228 |
| | KGE | 0.59844 | 0.600371 | 0.605007 | 0.612858 |
| | RMSE | 1.4434 | 1.43416 | 1.43009 | 1.38042 |
| | PBIAS | -0.547128 | -0.687947 | -0.865748 | -0.543918 |
| | lowRMSE | 0.118989 | 0.120228 | 0.115004 | 0.117728 |
| | highRMSE | 5.03277 | 5.02434 | 4.84415 | 4.74281 |
| | midRMSE | 0.462923 | 0.455257 | 0.453912 | 0.449598 |
| δHBV | NSE | 0.672839 | 0.644732 | 0.661231 | 0.684685 |
| | KGE | 0.653841 | 0.65646 | 0.6515 | 0.66205 |
| | RMSE | 1.43224 | 1.50803 | 1.48604 | 1.43376 |
| | PBIAS | 0.564363 | 1.55134 | -0.156553 | 0.956961 |
| | lowRMSE | 0.0975783 | 0.0984076 | 0.100773 | 0.100807 |
| | highRMSE | 4.83843 | 4.81176 | 4.72529 | 4.71255 |
| | midRMSE | 0.447828 | 0.431252 | 0.433688 | 0.432018 |
| LSTM+δHBV | NSE | 0.685032 | 0.680872 | 0.679321 | 0.700814 |
| | KGE | 0.638788 | 0.647826 | 0.646782 | 0.649999 |
| | RMSE | 1.35303 | 1.3873 | 1.36795 | 1.3185 |
| | PBIAS | -0.0150729 | 0.406127 | -0.135091 | -0.0232668 |





| | lowRMSE | 0.103284 | 0.101814 | 0.104528 | 0.102916 |
|---|---|---|---|---|---|
| | highRMSE | 4.80178 | 4.72583 | 4.70024 | 4.70713 |
| | midRMSE | 0.426819 | 0.411727 | 0.41573 | 0.41081 |

966

967



Table D4. *Median NSE, KGE, RMSE, PBIAS, and RMSE values under low (lowRMSE), high (highRMSE), and middle (midRMSE) flows based on 531 basins under the temporal, PUB, and PUR tests of $LSTM^{multi}$, $(LSTM + \delta HBV)^{123} + LSTM^{multi}$, their "seed" version, and $(LSTM + \delta HBV)^{123}_{seed}$.*

| Test | Metric | $LSTM^{multi}$ | $(LSTM + \delta HBV)^{123} + LSTM^{multi}$ |
|------|--------|----------------|---------------------------------------------|
| Temporal | NSE | 0.797448 | 0.82321 |
| | KGE | 0.811064 | 0.810248 |
| | RMSE | 1.05987 | 0.983168 |
| | PBIAS | 3.95241 | 4.08594 |
| | lowRMSE | 0.056221 | 0.05702 |
| | highRMSE | 2.7089 | 2.58881 |
| | midRMSE | 0.183526 | 0.192442 |
| PUB | NSE | 0.750605 | 0.782727 |
| | KGE | 0.71469 | 0.734731 |
| | RMSE | 1.20586 | 1.11509 |
| | PBIAS | 0.475674 | 0.706777 |
| | lowRMSE | 0.0861127 | 0.0836 |
| | highRMSE | 4.13615 | 3.83009 |
| | midRMSE | 0.347562 | 0.326814 |
| PUR | NSE | 0.623755 | 0.68923 |
| | KGE | 0.593757 | 0.633971 |
| | RMSE | 1.47379 | 1.31221 |



| PBIAS | -2.6737 | -1.38119 |
|---|---|---|
| lowRMSE | 0.112434 | 0.107646 |
| highRMSE | 4.98202 | 4.59232 |
| midRMSE | 0.501807 | 0.436811 |




*Table D4 (continued). Median NSE, KGE, RMSE, PBIAS, and RMSE values under low*
*(lowRMSE), high (highRMSE), and middle (midRMSE) flows based on 531 basins under the*
*temporal, PUB, and PUR tests of $LSTM^{multi}$, $(LSTM + \delta HBV)^{123} + LSTM^{multi}$, their*
*"seed" version, and $(LSTM + \delta HBV)^{123}_{seed}$.*

| Test | Metric | $(LSTM + \delta HBV)^{123}_{seed}$ | $LSTM^{multi}_{seed}$ | $(LSTM + \delta HBV)^{123}_{seed} + LSTM^{multi}_{seed}$ |
|------|--------|-----|-----|-----|
| Temporal | NSE | 0.821444 | 0.81992 | 0.829385 |
| | KGE | 0.795317 | 0.82078 | 0.812581 |
| | RMSE | 0.99455 | 1.00908 | 0.967779 |
| | PBIAS | 3.99009 | 4.09469 | 4.08882 |
| | lowRMSE | 0.059782 | 0.057346 | 0.057015 |
| | highRMSE | 2.7279 | 2.62815 | 2.58384 |
| | midRMSE | 0.209943 | 0.183656 | 0.195557 |
| PUB | NSE | 0.793673 | 0.781175 | 0.790921 |
| | KGE | 0.726188 | 0.736191 | 0.739284 |
| | RMSE | 1.12957 | 1.13079 | 1.09176 |
| | PBIAS | 0.370674 | 1.13671 | 0.869057 |
| | lowRMSE | 0.083423 | 0.084038 | 0.085728 |
| | highRMSE | 3.89363 | 3.93473 | 3.79505 |
| | midRMSE | 0.323045 | 0.329772 | 0.325627 |
| PUR | NSE | 0.705154 | 0.665723 | 0.701504 |
| | KGE | 0.651538 | 0.614649 | 0.64373 |
| | RMSE | 1.30377 | 1.3727 | 1.2851 |



| | PBIAS | -0.283645 | -2.74069 | -1.39149 |
|---|---|---|---|---|
| | lowRMSE | 0.100525 | 0.111229 | 0.108121 |
| | highRMSE | 4.74889 | 4.88127 | 4.58344 |
| | midRMSE | 0.406797 | 0.473783 | 0.432447 |







*Table D5. Median NSE values based on ten different random seeds during the temporal test.*
*Each number (1 through 10) represents metric values calculated for an individual simulation*
*based on only one random seed. "Seed" indicates metric values calculated by averages of*
*these ten simulations based on different random seeds, while "mean" denotes the average of*
*metrics from 1-10 individual simulations (visualized in Figure C1).*

| Number | $LSTM^{multi}$ | $(LSTM + \delta HBV)^{123}$ | $(LSTM + \delta HBV)^{123} + LSTM^{multi}$ |
|--------|---------|-------------------|--------------------------------|
| 1 | 0.797742 | 0.818436 | 0.82315 |
| 2 | 0.795312 | 0.820188 | 0.823559 |
| 3 | 0.799291 | 0.818097 | 0.822922 |
| 4 | 0.796388 | 0.818251 | 0.821791 |
| 5 | 0.791192 | 0.818285 | 0.820132 |
| 6 | 0.795691 | 0.81966 | 0.823268 |
| 7 | 0.795912 | 0.821511 | 0.82352 |
| 8 | 0.796625 | 0.81831 | 0.825204 |
| 9 | 0.794062 | 0.804959 | 0.816497 |
| 10 | 0.796066 | 0.817122 | 0.82169 |
| Seed | 0.82425 | 0.822528 | 0.832197 |
| Mean | 0.795828 | 0.817482 | 0.822173 |






**Appendix E: Intuitive visualization of the relative contributions of ensemble members based on optimized weights**

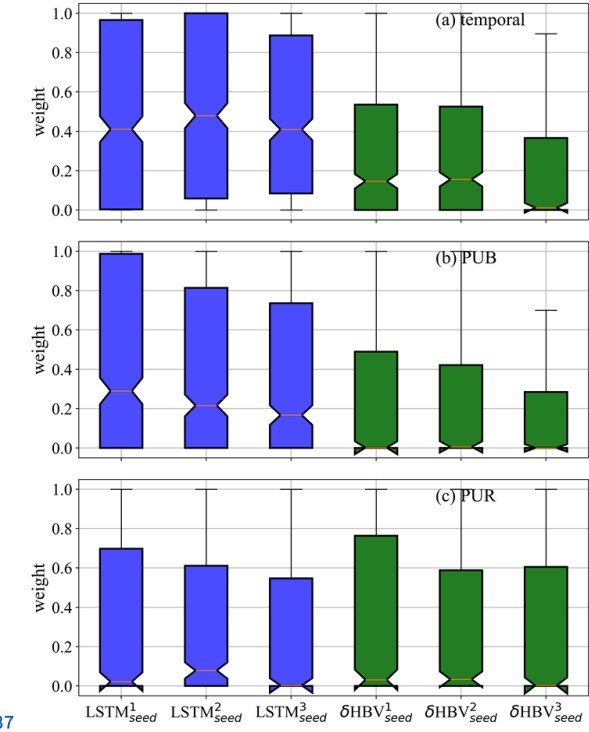

*Figure E1. Weights of six components across 531 basins, estimated basin-by-basin using a genetic algorithm based on streamflow observations during the test periods. The weights are normalized by the maximum weight within each ensemble group. These weights are used exclusively for qualitatively analyzing the relative contributions of different ensemble members, with higher values indicating larger relative contributions.*

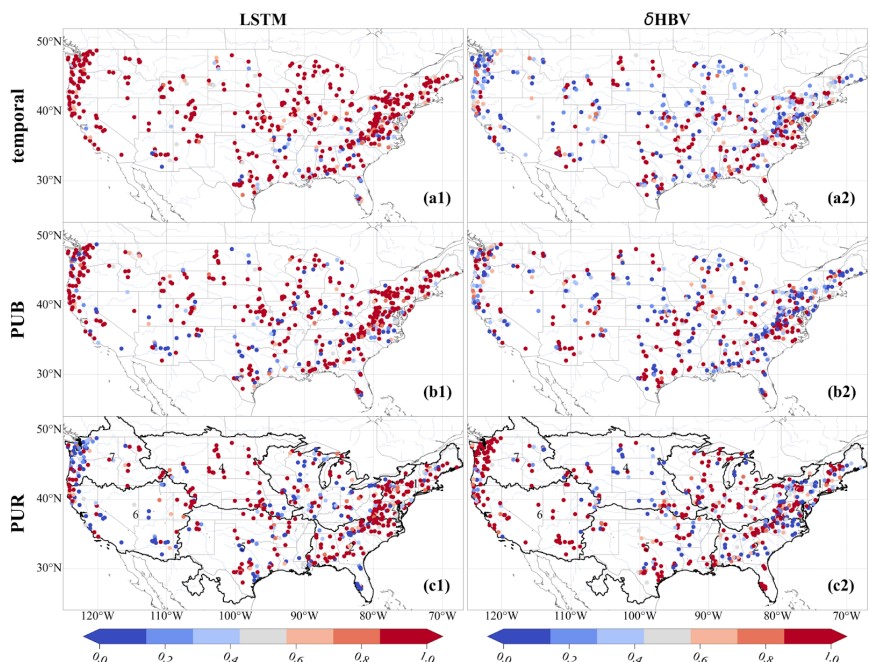

*Figure E2. Spatial distributions of weights of the LSTM and δHBV models, estimated by a genetic algorithm based on streamflow observations during the test periods. The weights are normalized by the maximum weight within each ensemble group. These weights are used exclusively for qualitatively analyzing the relative contributions of different ensemble members, with higher values indicating larger relative contributions.*



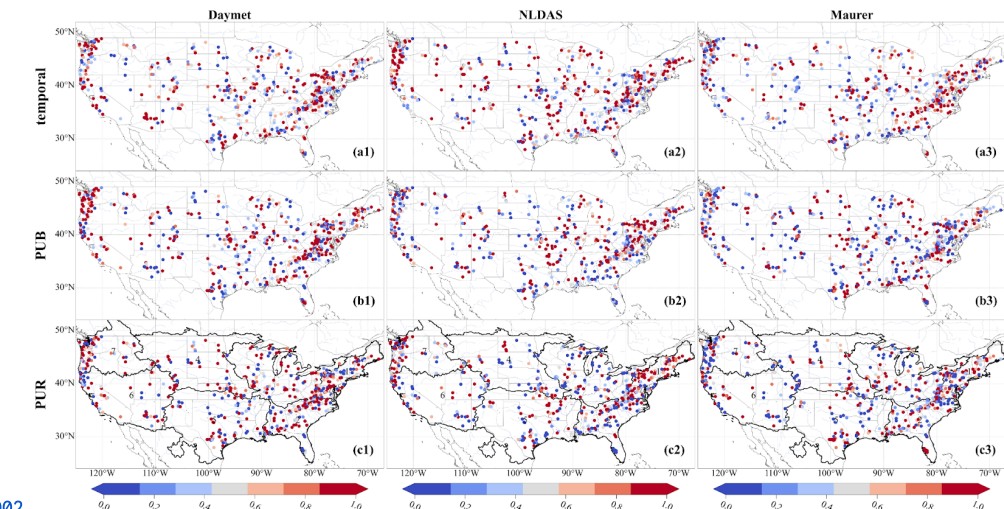

*Figure E3. Spatial distributions of weights of the Daymet, NLDAS, and Maurer meteorological forcing datasets, estimated by a genetic algorithm based on streamflow observations during the test periods. The weights are normalized by the maximum weight within each ensemble group. These weights are used exclusively for qualitatively analyzing the relative contributions of different ensemble members, with higher values indicating larger relative contributions.*



*Table E1. Comparisons of metric values between averaged ensemble simulations and*
*optimized weighted simulations, estimated using a genetic algorithm based on streamflow*
*observations during the test periods. The results highlight the potential for further*
*improvements in ensemble simulations.*

|  | Temporal | Averaged | Optimized weighted |
|---|---|---|---|
| Temporal | NSE | 0.821443912 | 0.844303212 |
|  | KGE | 0.795317495 | 0.829996445 |
|  | RMSE | 0.994550082 | 0.920954559 |
|  | PBIAS | 3.990094591 | 3.252278013 |
|  | lowRMSE | 0.059781616 | 0.057137161 |
|  | highRMSE | 2.72790133 | 2.451194907 |
|  | midRMSE | 0.20994263 | 0.183127162 |
| PUB | NSE | 0.793673 | 0.842396015 |
|  | KGE | 0.726188 | 0.79571295 |
|  | RMSE | 1.12957 | 0.987170488 |
|  | PBIAS | 0.370674 | 1.023040859 |
|  | lowRMSE | 0.0834234 | 0.079807878 |
|  | highRMSE | 3.89363 | 3.030715903 |
|  | midRMSE | 0.323045 | 0.285110115 |
| PUR | NSE | 0.705154 | 0.790796063 |
|  | KGE | 0.651538 | 0.746396324 |
|  | RMSE | 1.30377 | 1.13058149 |
|  | PBIAS | -0.283645 | 0.273698787 |
|  | lowRMSE | 0.100525 | 0.093595304 |
|  | highRMSE | 4.74889 | 3.665495069 |





| | midRMSE | 0.406797 | 0.351694421 |
|---|---------|----------|-------------|
