# Peer review of "Ensembling Differentiable Process-based and Data-driven Models with"

_EGUsphere, 2025_

## Referee Comment (RC1)

**Review of "Ensembling Differentiable Process-based and Data-driven Models with Meteorological Forcing Datasets to Advance Streamflow Simulation"**

**General comments**

This paper comprehensible evaluates the performance of different ensembles based on LSTM and δHBV models, and three different forcing datasets, across CAMELS catchments. The ensembles are evaluated in terms of a temporal test, a prediction in ungauged basins test (PUB), and a prediction in ungauged regions test (PUR). The main conclusion is that the data-driven LSTM and process-based δHBV ensemble improves NSE, particularly for PUB and PUR tests.

Overall, the manuscript is well-structured and clearly conveys its main point. Please find below some comments and suggestions.

**Specific comments**

1. L150 and L240-L244: Please explicitly indicate which features (static and dynamic) are used by the LSTM model or at least refer to Appendix C. Are static characteristics of the catchment also used during the PUB and PUR tests? Is it the case that for PUB the model does not use previous streamflow observations to generate the predictions? Or does PUB only refer to the model being tested at basins not used during training?

2. Table 2: Is it possible to draw any conclusions about the skill of the models to extrapolate to a warmer climate based on the temporal test? I assume that the period 1995-2010 is warmer than 1980-1995.

3. Table 2: What would happen if you were to train δHBV using only the same 531-basin subset as for the LSTM instead of the 671 basins?

4. It could be useful to also provide similar plots to Fig. 3 in the appendix where $\delta HBV^2$ and $\delta HBV^3$ are used instead of $\delta HBV^1$.

5. Fig. 4 and L439-L454: Can you expand on potential reasons for the lower model-skill in midwestern and western basins? Is human management of streamflow an important factor here, despite being CAMELS basins?

6. Fig. 5: Please clarify. L399-L401 says there is a small difference when using 3 or 10 seeds, but Fig. 5 shows the difference between individual seeds and using 10 seeds. It is interesting to note in Fig. 5 that $LSTM^{multi}$ with 10 seeds achieves a similar skill as $(LSTM + \delta HBV)^{123}$ at least for the temporal test. This could also be an important conclusion.

7. Fig. 6: Suggestion to have LSTM models with shades of one color and δHBV models with shades of a different color to better highlight the differences between LSTM and δHBV mentioned in L461-L464.

8. L467-L469: Suggestion to extend the sentence to clarify what is meant here.

**Minor comments and technical corrections**

1. L13: Replace "while" for "however".

2. L20: Suggest deleting "utilized in two ways". Here it just raises the question which two ways? Also in L526 it would be good to explicitly mention the "two ways".

3. L177-L179: Are all modifications to δHBV1.0 of similar importance? Or can it be said which of them are more important?

4. Table 2: Isn't there more recent data for PUB and PUR? Why are they trained only until 1999?

5. Figure B1. Xlabel on right panels should be temperature (C), correct?

---

## Author Comment (AC1)

*Reviewer #1*
*General comments*
*This paper comprehensible evaluates the performance of different ensembles based on LSTM and HBV models, and three different forcing datasets, across CAMELS catchments. The ensembles are evaluated in terms of a temporal test, a prediction in ungauged basins test (PUB), and a prediction in ungauged regions test (PUR). The main conclusion is that the data-driven LSTM and process-based HBV ensemble improves NSE, particularly for PUB and PUR tests.*

*Overall, the manuscript is well-structured and clearly conveys its main point. Please find below some comments and suggestions.*

Thanks for the positive comments

*Specific comments*
*1. L150 and L240-L244: Please explicitly indicate which features (static and dynamic) are used by the LSTM model or at least refer to Appendix C. Are static characteristics of the catchment also used during the PUB and PUR tests? Is it the case that for PUB the model does not use previous streamflow observations to generate the predictions? Or does PUB only refer to the model being tested at basins not used during training?*

Regarding the question "*Please explicitly indicate which features (static and dynamic) are used by the LSTM model or at least refer to Appendix C*", thanks for your suggestions. The static and dynamic attributes for LSTM are utilized as the same as Kratzert's studies (Kratzert et al., 2022), as shown in Table R1. We considered revising Table C1 of the original manuscript to specify the inputs for LSTM and $\delta$ HBV, respectively.

Regarding the question "*Are static characteristics of the catchment also used during the PUB and PUR tests?*",  yes. These static characteristics of the catchment are also used during the PUB and PUR tests.

Regarding the question "*Is it the case that for PUB the model does not use previous streamflow observations to generate the predictions?*", yes. For all three kinds of tests, streamflow observations are not included in the inputs and are only used to calibrate the model.

Regarding the question "*does PUB only refer to the model being tested at basins not used during training?*" -- yes, exactly. As described in Section 2.5 of the manuscript, we first divided all basins into 10 subsets. The model was then trained and evaluated over 10 rounds, each time holding out one subset for testing while using the remaining basins for training. After completing all rounds, the test results from all basins were concatenated to evaluate overall performance. Therefore, in each round of evaluation, the test basins were strictly excluded from the training process.

Table R1. Full names for the abbreviations of dynamic data (all but streamflow are "forcings") and static basin attributes used as the LSTM model inputs and outputs. All variables and their values are provided in the CAMELS dataset (Addor et al., 2017) except for the NLDAS and Maurer daily temperature extrema, which are from Kratzert et al. (2021).

| Type | Abbreviation | Full name | Unit |
| --- | --- | --- | --- |

| | | | |
|---|---|---|---|
| **Dynamic forcings** | prcp | Precipitation | mm/day |
| | tmax | Maximum air temperature | °C |
| | tmin | Minimum air temperature | °C |
| | srad | Shortwave radiation | W/m$^2$ |
| | vp | Water vapor pressure | pa |
| | q | Streamflow normalized by basin area (q_vol / area_gages2) | mm/day |
| **Static basin attributes** | p_mean | Mean daily precipitation | mm/day |
| | pet_mean | Mean daily potential evapotranspiration | mm/day |
| | frac_snow | Fraction of precipitation falling as snow | - |
| | aridity | Rate of mean values of potential evapotranspiration and precipitation | - |
| | high_prec_freq | Frequency of high precipitation days | days/year |
| | high_prec_dur | Average duration of high precipitation events | days |
| | low_prec_freq | Frequency of dry days | days/year |
| | low_prec_dur | Average duration of dry periods | days |
| | elev_mean | Catchment mean elevation | m |
| | slope_mean | Catchment mean slope | m/km |
| | area_gages2 | Catchment area (GAGES-II estimate) | km$^2$ |
| | frac_forest | Fraction of catchment area having land cover identified as forest | - |
| | lai_max | Maximum monthly mean of the leaf area index | - |
| | lai_diff | Difference between the maximum and minimum monthly mean of the leaf area index | - |
| | gvf_max | Maximum monthly mean of the green vegetation | - |
| | gvf_diff | Difference between the maximum and minimum monthly mean of the green vegetation fraction | - |
| | soil_depth_pelletier | Depth to bedrock | m |
| | soil_depth_statsgso | Soil depth | m |
| | soil_porosity | Volumetric soil porosity | - |

| | soil_conductivity | Saturated hydraulic conductivity | cm/hr |
|---|---|---|---|
| | max_water_content | Maximum water content | m |
| | sand_frac | Fraction of soil which is sand | - |
| | silt_frac | Fraction of soil which is silt | - |
| | clay_frac | Fraction of soil which is clay | - |
| | carbonate_rocks_frac | Fraction of the catchment area as carbonate sedimentary rocks | - |
| | geol_permeability | Subsurface permeability | $m^2$ |

*2. Table 2: Is it possible to draw any conclusions about the skill of the models to extrapolate to a warmer climate based on the temporal test? I assume that the period 1995-2010 is warmer than 1980-1995.*

Thanks for the suggestions. We have compared the temperature changes from the training to the test periods, as shown in Figure R1. The results show that most basins are getting warmer, and these models can also get satisfactory performances, which indicates that these models can extrapolate at least modestly their performances under a warmer climate. Note that it is an often asked question for purely data-driven models like LSTM to lose accuracy if the drift is stronger. The issue is that there was a lack of a "substantially warmed" real dataset to assess this behavior. Previous benchmarks suggest the 15-year-scale trend to be accurately captured in temporal test by LSTM, and more poorly captured for PUR (Feng et al., 2023). We will add some sentences in the revised manuscript to describe this finding.

[Figure]

Figure R1. Boxplot of relative temperature differences between the test and training periods, calculated as (Test − Training) / Training. Each box represents the distribution of normalized temperature changes across basins for a specific meteorological forcing dataset: Daymet, NLDAS, and Maurer. Positive values indicate warming in the test period relative to the training period

*3. Table 2: What would happen if you were to train HBV using only the same 531-basin subset as for the LSTM instead of the 671 basins?*

Thanks for the question. We tested this by conducting the temporal experiments using only the 531-basin subset for the δHBV model. The results were largely similar to those obtained using all 671 basins shown in Figures R2-R3, indicating that the impact of reducing the training set size is limited in this context. We will incorporate some discussion about this in the revised manuscript.

We believe that training the δHBV model on the full 671-basin dataset is still beneficial, as the physical constraints inherent in the model allow it to make more effective use of available data, even when data quality is somewhat limited. That said, the added value from including the additional 140 basins appears to be marginal, and the choice of training on 531 versus 671 basins does not substantially affect the overall model performance.

[Figure]

Figure R2. Comparison of δHBV simulations trained on 671 versus 531 basins across performance metrics (a)–(g), with test cases distinguished by varying x-axis labels

[Figure]

Figure R3. Comparison of NSE spatial distributions for δHBV models trained on 671 versus 531 basins across different meteorological forcing datasets

*4. It could be useful to also provide similar plots to Fig. 3 in the appendix where HBV2 and HBV3 are used instead of HBV1.*

Thanks for the suggestions. We have plotted Figures R4 and R5 based on δHBV2 and δHBV3, and they show similar results, consistent with the conclusions. We will revise the manuscript accordingly.

[Figure]

Figure R4. Scatter plots comparing the performance differences between hydrological models for the basins where LSTM outperformed δHBV (the basins where δHBV outperformed are not shown in this plot). The x-axis represents the NSE differences between $LSTM^2$ and $\delta HBV^2$ ($LSTM^2$ - $\delta HBV^2$), while the y-axis shows the NSE differences between $\delta HBV^{123}$ and $\delta HBV^2$ ($\delta HBV^{123}$ - $\delta HBV^2$). Points are color-coded according to the NSE values of $\delta HBV^2$. The correlation coefficient (CORR) and p values between x-axis values and y-axis values, along with the median NSE value of $\delta HBV^2$ ($NSE_{med}$) on these basins are also noted.

[Figure]

Figure R5. Scatter plots comparing the performance differences between hydrological models for the basins where LSTM outperformed δHBV (the basins where δHBV outperformed are not shown in this plot). The x-axis represents the NSE differences between $LSTM^3$ and $\delta HBV^3$ ($LSTM^3$ - $\delta HBV^3$), while the y-axis shows the NSE differences between $\delta HBV^{123}$ and $\delta HBV^3$ ($\delta HBV^{123}$ - $\delta HBV^3$). Points are color-coded according to the NSE values of $\delta HBV^3$. The correlation coefficient (CORR) and p values between x-axis values and y-axis values, along with the median NSE value of $\delta HBV^3$ ($NSE_{med}$) on these basins are also noted.

*5. Fig. 4 and L439-L454: Can you expand on potential reasons for the lower model-skill in midwestern and western basins? Is human management of streamflow an important factor here, despite being CAMELS basins?*

Thanks for the suggestions. We have directly compared the spatial patterns of performance with the spatial distributions of evaporation and the dam number. We found that over the midwestern and western CONUS, there are also high evaporation climate conditions (Figure 2 of (Heidari et al., 2020), shown here as Figure R6) and a large number of basins (Figure 1 of (Ryan Bellmore et al., 2017), shown here as Figure R7), which tend to have complex water use processes that cannot be simulated via the models (Figure 11 of (Wada et al., 2016), shown here as Figure R8). All these factors indicate that anthropogenetic influence can be an important factor that causes the model to perform poorly. And these factors have been implicitly expressed in lines 486-490 in section 3.3 of the original manuscript. Based on your suggestions, these sentences will be revised to be clearer.

[Figure]

Figure R6. Maps of current (a) aridity index and (b) evaporative index for the baseline period (1986–2015) from (Heidari et al., 2020).

[Figure]

Figure R7. Distribution of dams in the contiguous U.S. from the corrected figure of (Ryan Bellmore et al., 2017).

[Figure]

Figure R8. Water Stress Index for 2010 calculated at a 6 min spatial resolution from (Wada et al., 2016)

*6. Fig. 5: Please clarify. L399-L401 says there is a small difference when using 3 or 10 seeds, but Fig. 5 shows the difference between individual seeds and using 10 seeds. It is interesting to note in Fig. 5 that $LSTM^{multi}$ with 10 seeds achieves a similar skill as $(LSTM + HBV)^{123}$ at least for the temporal test. This could also be an important conclusion.*

Thanks for the comments. It is true that LSTM, after ensembling different random seeds, increases much higher compared with the individual component and other ensemble strategies. But here, we want to show that there are smaller differences between ensembling 3 random seeds and ensembling 10 seeds. Specific metric values of the LSTM model with different random seeds can be found in Table D4 and Table D5 (0.81992 (3 seeds) v.s. 0.824 (10 seeds)). We also implicitly describe the boost performance improvement of $LSTM^{multi}_{seed}$ and owe it to the instability of LSTM simulations in the original manuscript as, "The

performance of $(LSTM + \delta HBV)^{123}$ ensemble proved more robust than $LSTM^{multi}$, with only a slight boost when we incorporated random seeds, i.e., $(LSTM + \delta HBV)^{123}_{seed}$." in lines 390-391. Based on your suggestions, we will add some sentences to make it clearer in the revised manuscript.

*7. Fig. 6: Suggestion to have LSTM models with shades of one color and HBV models with shades of a different color to better highlight the differences between LSTM and HBV mentioned in L461-L464.*

Thanks for the suggestions. We replotted the figure accordingly as shown in Figure R9 here.

[Figure]

Figure R9. Comparisons between multi-basin-averaged streamflow observations and simulations across 531 basins. The time series points are displayed at four-day intervals for clarity and conciseness. Ensemble members based on the same model (LSTM or δHBV) but driven by different forcing datasets are shown in the same color to highlight the differences between models more clearly.

*8. L467-L469: Suggestion to extend the sentence to clarify what is meant here.*

Thanks for the suggestion. We will revise the original sentence

"This highlights the critical importance of comprehensive training for each ensemble member to enable the development of distinct characteristics in their streamflow simulations, ultimately enhancing ensemble performance."

as,

"This highlights the critical importance of comprehensive training for each ensemble member, including diverse forcing inputs, full-period model calibration, and rigorous hyperparameter tuning, to ensure that each member develops distinct simulation behaviors. These differences allow the ensemble to better represent a range of hydrological responses, particularly under extreme or uncertain conditions. By capturing complementary strengths and compensating for individual weaknesses, such well-trained ensemble members collectively enhance the robustness and accuracy of streamflow simulations."

*Minor comments and technical corrections*

*1. L13: Replace "while" for "however".*

Thanks, it will be revised.

*2. L20: Suggest deleting "utilized in two ways". Here it just raises the question which two ways? Also in L526 it would be good to explicitly mention the "two ways".*

Thanks for the suggestions. We will delete the "utilized in two ways" in the Abstract section.
As for "*Also in L526 it would be good to explicitly mention the "two ways".*" in the conclusion section, we will revise the original words as,
"Three meteorological forcing datasets (Daymet, NLDAS, and Maurer) are employed to fully capture the characteristics of the two models. Their applications are also tested in two distinct ways: (1) by feeding all diverse forcing datasets simultaneously into a single LSTM model, and (2) by ensembling the outputs of multiple LSTM models, each trained separately using a single forcing dataset."

*3. L177-L179: Are all modifications to HBV1.0 of similar importance? Or can it be said which of them are more important?*

Thanks for the insightful comment. We are very cautious to make modifications, and to determine these modifications, we have evaluated various structural changes across multiple studies using diverse datasets. Each modification targets specific aspects of model improvement, and most contribute significantly to overall performance.

To address high-flow simulation challenges, we implemented three key modifications: the use of three dynamic parameters ($\gamma$, $\beta$, $\theta_{K0}$) during training and testing periods; the removal of log-transform normalization for precipitation; and the adoption of the normalized squared-error loss function.

Our recent study (Song et al., 2024) shows that $\delta$HBV1.1p with three dynamic parameters ($\gamma$, $\beta$, $\theta_{K0}$) outperforms the two-parameter version ($\gamma$, $\beta$). The dynamic shape coefficient ($\beta$) and evapotranspiration coefficient ($\gamma$) capture the nonlinear relationships between surface soil moisture and effective rainfall, as

well as evapotranspiration. The dynamic $\theta_{K0}$ parameter reflects variable water release rates influenced by changing groundwater levels, bank and wetland storage, and other factors. By remaining small during low-flow periods and increasing during peak-flow events, dynamic $\theta_{K0}$ allows the upper soil layer to retain more moisture before extreme events, thereby enhancing peak-flow contributions..

The elimination of log-transform normalization for precipitation, paired with the adoption of the normalized squared-error (NSE) loss function, synergistically enhances model performance. By removing log-transform normalization, the model becomes more sensitive to high precipitation events, thus better capturing high-flow conditions. Simultaneously, the NSE loss function amplifies the impact of significant deviations in peak flows, further improving the model's ability to predict high-flow events effectively (Frame et al., 2022; Kratzert et al., 2021; Song et al., 2025a, b; Wilbrand et al., 2023).

In contrast, maintaining dynamic parameters during warm-up periods offers marginal benefits while increasing computational costs. However, it provides a more realistic representation and mitigates potential uncertainties from initial conditions.

Based on your feedback, we will revise the relevant sections with more details for clarity and precision.

*4. Table 2: Isn't there more recent data for PUB and PUR? Why are they trained only until 1999?*

Thanks for the question. All three tests (temporal, PUB, and PUR) are conducted using the same underlying dataset. However, due to differences in testing strategies, the computational cost for PUB and PUR is significantly higher than for the temporal test. Specifically, each complete evaluation requires 10 runs for PUB and 7 runs for PUR. Based on prior studies (Feng et al., 2021, 2023; Kratzert et al., 2019) and to balance computational efficiency with the objectives of our analysis, we limited the training period to data up to 1999. This choice allows us to preserve the core evaluation goals while keeping the computational demand manageable.

*5. Figure B1. Xlabel on right panels should be temperature (C), correct?*

Thanks for pointing it out. It has been replotted (Figure R10) and will be added in the revised manuscript.

[Figure]

Figure R10. *Probability density distributions of precipitation and temperature across three meteorological forcing datasets.*

**References:**

Addor, N., Newman, A. J., Mizukami, N., and Clark, M. P.: The CAMELS data set: catchment attributes and meteorology for large-sample studies, Hydrol. Earth Syst. Sci., 21, 5293–5313, https://doi.org/10.5194/hess-21-5293-2017, 2017.

Feng, D., Lawson, K., and Shen, C.: Mitigating prediction error of deep learning streamflow models in large data-sparse regions with ensemble modeling and soft data, Geophysical Research Letters, 48, e2021GL092999, https://doi.org/10.1029/2021GL092999, 2021.

Feng, D., Beck, H., Lawson, K., and Shen, C.: The suitability of differentiable, physics-informed machine learning hydrologic models for ungauged regions and climate change impact assessment, Hydrology and Earth System Sciences, 27, 2357–2373, https://doi.org/10.5194/hess-27-2357-2023, 2023.

Frame, J. M., Kratzert, F., Klotz, D., Gauch, M., Shalev, G., Gilon, O., Qualls, L. M., Gupta, H. V., and Nearing, G. S.: Deep learning rainfall–runoff predictions of extreme events, Hydrology and Earth System Sciences, 26, 3377–3392, https://doi.org/10.5194/hess-26-3377-2022, 2022.

Heidari, H., Arabi, M., Warziniack, T., and Kao, S.-C.: Assessing shifts in regional hydroclimatic conditions of U.S. river basins in response to climate change over the 21st century, Earth's Future, 8, e2020EF001657, https://doi.org/10.1029/2020EF001657, 2020.

Kratzert, F., Klotz, D., Herrnegger, M., Sampson, A. K., Hochreiter, S., and Nearing, G. S.: Toward improved predictions in ungauged basins: Exploiting the power of machine learning, Water Resources Research, 55, 11344–11354, https://doi.org/10/gg4ck8, 2019.

Kratzert, F., Klotz, D., Hochreiter, S., and Nearing, G. S.: A note on leveraging synergy in multiple meteorological data sets with deep learning for rainfall–runoff modeling, Hydrology and Earth System Sciences, 25, 2685–2703, https://doi.org/10.5194/hess-25-2685-2021, 2021.

Kratzert, F., Gauch, M., Nearing, G., and Klotz, D.: NeuralHydrology — A Python library for Deep Learning research in hydrology, , https://doi.org/10.5281/zenodo.6326394, 2022.

Ryan Bellmore, J., Duda, J. J., Craig, L. S., Greene, S. L., Torgersen, C. E., Collins, M. J., and Vittum, K.: Status and trends of dam removal research in the United States, WIREs Water, 4, e1164, https://doi.org/10.1002/wat2.1164, 2017.

Song, Y., Knoben, W. J. M., Clark, M. P., Feng, D., Lawson, K., Sawadekar, K., and Shen, C.: When ancient numerical demons meet physics-informed machine learning: adjoint-based gradients for implicit differentiable modeling, Hydrology and Earth System Sciences, 28, 3051–3077, https://doi.org/10.5194/hess-28-3051-2024, 2024.

Song, Y., Bindas, T., Shen, C., Ji, H., Knoben, W. J. M., Lonzarich, L., Clark, M. P., Liu, J., van Werkhoven, K., Lamont, S., Denno, M., Pan, M., Yang, Y., Rapp, J., Kumar, M., Rahmani, F., Thébault, C., Adkins, R., Halgren, J., Patel, T., Patel, A., Sawadekar, K. A., and Lawson, K.: High-resolution national-scale water modeling is enhanced by multiscale differentiable physics-informed machine learning, Water Resour. Res., 61, e2024WR038928, https://doi.org/10.1029/2024WR038928, 2025a.

Song, Y., Sawadekar, K., Frame, J. M., Pan, M., Clark, M., Knoben, W. J. M., Wood, A. W., Lawson, K. E., Patel, T., and Shen, C.: Physics-informed, differentiable hydrologic models for capturing unseen extreme events, https://doi.org/10.22541/essoar.172304428.82707157/v2, 2025b.

Wada, Y., de Graaf, I. E. M., and van Beek, L. P. H.: High-resolution modeling of human and climate impacts on global water resources, Journal of Advances in Modeling Earth Systems, 8, 735–763, https://doi.org/10/f8wgpv, 2016.

Wilbrand, K., Taormina, R., ten Veldhuis, M.-C., Visser, M., Hrachowitz, M., Nuttall, J., and Dahm, R.: Predicting streamflow with LSTM networks using global datasets, Front. Water, 5, https://doi.org/10.3389/frwa.2023.1166124, 2023.

---

## Author Comment (AC2)

*Reviewer #2*
*The manuscript presents an innovative ensemble strategy that combines a differentiable process-based model (δHBV) with a data-driven Long Short-Term Memory (LSTM) model, further diversified through the application of multiple meteorological forcing datasets. The approach is evaluated across a wide range of generalization scenarios (temporal extrapolation, PUB, and PUR) using the CAMELS dataset. Although the paper is well-written and the main ideas are clearly communicated, it would benefit from additional details in the methods and a deeper discussion of model complementarity and limitations.*

Thanks for the positive comments and constructive suggestions. We will revise the manuscript accordingly. Please find our point-by-point responses below.

*Strength*

*The proposed ensemble framework is conceptually strong and offers a well-justified combination of complementary data and algorithmic modeling paradigms.*
*The study is well evaluated across well-defined training protocols and temporal-spatial splits, which improves confidence in its generalizability.*
*The use of multiple data sources for meteorological forcings addresses input uncertainty better than traditional single-source modeling.*
*The results, specifically the finding that δHBV improves spatial generalization, have clear implications for prediction in ungauged regions.*

Thanks for the positive comments.

*Weakness*

*Interpretability: While the δHBV model's performance is shown to be beneficial in spatial generalization, the underlying reasons for this complementarity (e.g., structural constraints, parameter smoothness) are not deeply explored. A discussion of how each model contributes to ensemble diversity would strengthen the scientific value of the work.*

We thank the reviewer for this insightful comment regarding model interpretability and the complementarity between LSTM and δHBV within the ensemble framework. We fully agree that a deeper understanding of the relative contributions of each model would enhance the scientific value of our study. Besides this, we also plan to dig deeper and examine cases where the errors of LSTM and dHBV cancel each other.

[Figure]

*Figure R1. Boxplots of the spread values of simulations based on LSTM, δHBV, and LSTM + δHBV with different meteorological forcings and random seeds across temporal, PUB, and PUR tests.*

[Figure]

*Figure R2. Spatial distributions of spread increase from δHBV and LSTM to the LSTM+δHBV ensemble across temporal, PUB, and PUR tests.*

[Figure]

*Figure R3. Distributions of observation–simulation pairs from LSTM and δHBV models along the 1:1 line across temporal, PUB, and PUR tests. Percentages of pairs lying above the 1:1 line for both models are also indicated.*

Since the benefits of the different ensemble members to the deterministic precision have been displayed in the original manuscript, we have conducted additional analyses in terms of ensemble variability as suggested. Specifically, we use the spread values (Li et al., 2021; Reichle and Koster, 2003), which are widely adopted to quantify ensemble variability, to further explore model complementarity. The spread value is calculated as follows,

$$Spread = \sqrt{\frac{1}{n}\frac{1}{r}\sum_{i=1}^{n}\sum_{j=1}^{r}(S_{i,j} - \overline{S_i})^2}$$

Where $n$ is the number of simulated days, r is the number of ensemble members, and S is the simulations of each ensemble member, $\overline{()}$ indicates the average of values.

Figure R1 presents the boxplots of spread values for ensemble simulations using random seed variations with LSTM, δHBV, and the combined LSTM + δHBV, across the temporal, PUB, and PUR test settings. We observe that the overall spread increases from temporal to PUB and PUR tests, reflecting growing uncertainty. Notably, δHBV consistently exhibits lower spread values than LSTM across all tests, indicating its higher stability. This aligns with our prior discussion: δHBV tends to constrain the learnable function space, thus having lower variability and potentially higher bias. This difference stems from their structural characteristics—δHBV is governed by more rigid physical constraints, which limit unrealistic dynamics and enhance stability, while LSTM is more flexible and capable of capturing patterns that may not be explicitly represented in physical models, such as human influences or unmodeled processes. The combination of both models (LSTM + δHBV) yields greater spread values, indicating enhanced ensemble diversity. This suggests that the two models offer complementary strengths—LSTM contributes flexibility and capacity to represent data-driven nuances, while δHBV anchors the ensemble with physically constrained behavior.

Figure R2 illustrates the spatial distributions of spread increase resulting from incorporating LSTM and δHBV, respectively, and further supports our previous analysis. Incorporating LSTM leads to an increase in spread values across all basins, reflecting its higher variability. In contrast, the δHBV model, characterized by stronger physical constraints and generally lower variability, results in a decrease in spread values for many basins. However, δHBV still contributes to a spread increase in most northern basins and gradually leads to spread increases in a larger number of basins across the CONUS. This suggests notable differences in simulated streamflow behavior between LSTM and δHBV, largely attributable to their distinct model structures. Figure R3 reveals relatively limited differences between the streamflow behaviors simulated by LSTM and δHBV, with LSTM generally producing higher streamflow estimates than δHBV. A more systematic investigation of these differences would be valuable in future studies.

Following the reviewer's suggestion, we will incorporate these analyses and discussions about the ensemble spread in the revised manuscript.

*Robustness and Sensitivity Analysis: The paper lacks an explicit assessment of how ensemble performance responds to errors or biases in the forcing datasets or uncertainty in model parameters. Including even a limited robustness analysis would improve confidence in the ensemble's reliability. Additionally, the authors should consider running one or two experiments to understand whether changing the size of the lookback window (i.e., the number of historical timesteps) for the LSTMs impacted the overall performance of the ensemble.*

Thank you for the suggestions. Based on them, we conducted several experiments using temporal tests to demonstrate the robustness of ensemble benefits under various factors, including precipitation errors, parameter uncertainties in the δHBV model, and hyperparameter uncertainties in the LSTM model.

Regarding sensitivity to the forcing datasets, we ran the δHBV and LSTM models under a temporal test, both without and with a precipitation error introduced by multiplying precipitation by 0.1, to examine differences across ensemble groups. The results, shown in Figures R4 and R5, indicate that although the performance of both LSTM and δHBV decreases when the precipitation error is introduced, the decrease is not substantial, demonstrating a certain degree of robustness to precipitation errors and some capacity of both models to adapt to such errors. Interestingly, LSTM and δHBV respond differently to this type of precipitation error: for LSTM, the error tends to reduce ensemble performance mainly under low and high flow regimes, whereas for δHBV, the reduction is more pronounced under low and middle flow regimes. These differences reflect the fact that LSTM does not need to respect mass balance and can adjust precipitation up or down internally, but has trouble learning the contrast, while δHBV needs to distort the low flow to capture the high flows. Despite these differences, the ensemble benefits remain significant and robust when comparing different ensemble groups and assessing the impact of precipitation errors.

[Figure]

*Figure R4. Simulation performance under the temporal test using the LSTM model with and without a precipitation error equal to 0.1 times the precipitation, compared across metrics (a)–(g).*

[Figure]

*Figure R5. Simulation performance under the temporal test using the δHBV model with and without a precipitation error equal to 0.1 times the precipitation, compared across metrics (a)–(g).*

Similar results are observed in cases investigating the effects of parameter uncertainties in δHBV (Figure R6) and hyperparameter uncertainties in LSTM (Figure R7). Regarding parameter uncertainties, we additionally ran a case using the δHBV model with fewer dynamic parameters—reducing the number from three in the benchmark case to two—by fixing the infiltration rate parameter K0 as static to assess the resulting performance changes, which may reduce δHBV's ability to represent dynamic water release processes influenced by changing groundwater levels, bank and wetland storages, and other factors (Song et al., 2025b). This leads to increased structural errors and decreased model performance. Nevertheless,

the contribution of δHBV to ensemble simulations remains robust, with ensemble benefits substantially outweighing the negative effects of parameter uncertainties.

[Figure]

*Figure R6. Simulation performance under the temporal test using the δHBV model with 3 and 2 dynamic parameters, compared across metrics (a)–(g).*

Regarding hyperparameter uncertainties in the LSTM model, we focus on a key hyperparameter: the lookback window size, as suggested. We treat this parameter as having physical significance related to the temporal period rather than a typical hyperparameter. Therefore, we fix the window size to one year (365 days) to capture a full annual cycle while accounting for interannual variability. To evaluate the impact of

different window lengths, we include two additional scenarios with 182 and 730 timesteps. As shown in Figure R7, the LSTM model with a 365-day window generally achieves better performance across most scenarios. However, compared to the overall benefits of the ensemble, this difference is not substantial, indicating the robustness of ensemble simulations to variations in this LSTM hyperparameter.

[Figure]

*Figure R7. Simulation performance under the temporal test using the δHBV model on the time steps of 365, 182, 730, compared across metrics (a)–(g).*

Although it is practically impossible to test the effects of all possible configurations on ensemble benefits, we expect these benefits to remain robust against other factors to some extent, based on the representative results presented. Following the suggestions, we will include these additional cases in the revised manuscript to further demonstrate the reliability of our ensemble simulations.

*Scalability and Practical Deployment: The manuscript does not address the computational or operational feasibility of deploying this ensemble framework in practice, especially over large domains or in real-time forecasting contexts. A short discussion (1-2 sentences) on this topic would add practical relevance.*

We appreciate the reviewer's suggestion to further discuss the computational and operational feasibility of deploying the ensemble framework. This point is partially addressed in Section 3.3 of the original manuscript, where we note:

"Moreover, ensemble simulations may face challenges when computational resources are limited and calculations are performed sequentially. However, we remain optimistic about these challenges, as the processes can be addressed by leveraging parallel computing with multiple GPUs, benefiting from ongoing advancements in computational power."

In response to the reviewer's comment, we plan to expand this discussion as follows:

[revised manuscript text omitted]

---

## Author Response (AR1)

We sincerely appreciate the editor's positive assessment and the constructive suggestions from both reviewers. In response, we have revised the manuscript accordingly, adding the requested clarifications and additional details as suggested.

In addition to these revisions, we have adjusted the model results using more order-consistent random seeds in the temporal tests. This change has almost no effect on model performance, further confirming the stability of the ensemble benefits, but improves the logical consistency of the study.

*Reviewer #1*
*General comments*
*This paper comprehensible evaluates the performance of different ensembles based on LSTM and HBV models, and three different forcing datasets, across CAMELS catchments. The ensembles are evaluated in terms of a temporal test, a prediction in ungauged basins test (PUB), and a prediction in ungauged regions test (PUR). The main conclusion is that the data-driven LSTM and process-based HBV ensemble improves NSE, particularly for PUB and PUR tests.*

*Overall, the manuscript is well-structured and clearly conveys its main point. Please find below some comments and suggestions.*

Thanks for the positive comments.

*Specific comments*
*1. L150 and L240-L244: Please explicitly indicate which features (static and dynamic) are used by the LSTM model or at least refer to Appendix C. Are static characteristics of the catchment also used during the PUB and PUR tests? Is it the case that for PUB the model does not use previous streamflow observations to generate the predictions? Or does PUB only refer to the model being tested at basins not used during training?*

Regarding the question "*Please explicitly indicate which features (static and dynamic) are used by the LSTM model or at least refer to Appendix C*", thanks for your suggestions. The static and dynamic attributes for LSTM are utilized as the same as Kratzert's studies (Kratzert et al., 2022), as shown in Table R1. We have revised Table C1 of the original manuscript to specify the inputs for LSTM and $\delta$HBV, respectively.

Regarding the question "*Are static characteristics of the catchment also used during the PUB and PUR tests?*", yes. These static characteristics of the catchment are also used during the PUB and PUR tests.

Regarding the question "*Is it the case that for PUB the model does not use previous streamflow observations to generate the predictions?*", yes. For all three kinds of tests, streamflow observations are not included in the inputs and are only used to calibrate the model.

Regarding the question "*does PUB only refer to the model being tested at basins not used during training?*" -- yes, exactly. As described in Section 2.5 of the manuscript, we first divided all basins into 10 subsets. The model was then trained and evaluated over 10 rounds, each time holding out one subset for testing while using the remaining basins for training. After completing all rounds, the test results from all basins were concatenated to evaluate overall performance. Therefore, in each round of evaluation, the test basins were strictly excluded from the training process.

Table R1. Full names for the abbreviations of dynamic data (all but streamflow are "forcings") and static basin attributes used as the LSTM model inputs and outputs. All variables and their values are provided in the CAMELS dataset *(Addor et al., 2017)* except for the NLDAS and Maurer daily temperature extrema, which are from Kratzert et al. *(2021)*.

| Type | Abbreviation | Full name | Unit |
|---|---|---|---|
| **Dynamic forcings** | prcp | Precipitation | mm/day |
| | tmax | Maximum air temperature | °C |
| | tmin | Minimum air temperature | °C |
| | srad | Shortwave radiation | $W/m^2$ |
| | vp | Water vapor pressure | pa |
| | q | Streamflow normalized by basin area (q_vol / area_gages2) | mm/day |
| **Static basin attributes** | p_mean | Mean daily precipitation | mm/day |
| | pet_mean | Mean daily potential evapotranspiration | mm/day |
| | frac_snow | Fraction of precipitation falling as snow | - |
| | aridity | Rate of mean values of potential evapotranspiration and precipitation | - |
| | high_prec_freq | Frequency of high precipitation days | days/year |
| | high_prec_dur | Average duration of high precipitation events | days |
| | low_prec_freq | Frequency of dry days | days/year |
| | low_prec_dur | Average duration of dry periods | days |
| | elev_mean | Catchment mean elevation | m |
| | slope_mean | Catchment mean slope | m/km |
| | area_gages2 | Catchment area (GAGES-II estimate) | $km^2$ |
| | frac_forest | Fraction of catchment area having land cover identified as forest | - |

| | | | |
|---|---|---|---|
| | lai_max | Maximum monthly mean of the leaf area index | - |
| | lai_diff | Difference between the maximum and minimum monthly mean of the leaf area index | - |
| | gvf_max | Maximum monthly mean of the green vegetation | - |
| | gvf_diff | Difference between the maximum and minimum monthly mean of the green vegetation fraction | - |
| | soil_depth_pelletier | Depth to bedrock | m |
| | soil_depth_statsgso | Soil depth | m |
| | soil_porosity | Volumetric soil porosity | - |
| | soil_conductivity | Saturated hydraulic conductivity | cm/hr |
| | max_water_content | Maximum water content | m |
| | sand_frac | Fraction of soil which is sand | - |
| | silt_frac | Fraction of soil which is silt | - |
| | clay_frac | Fraction of soil which is clay | - |
| | carbonate_rocks_frac | Fraction of the catchment area as carbonate sedimentary rocks | - |
| | geol_permeability | Subsurface permeability | $m^2$ |

*2. Table 2: Is it possible to draw any conclusions about the skill of the models to extrapolate to a warmer climate based on the temporal test? I assume that the period 1995-2010 is warmer than 1980-1995.*

Thanks for the suggestions. We have compared the temperature changes from the training to the test periods, as shown in Figure R1. The results show that most basins are getting warmer, and these models can also get satisfactory performances, which indicates that these models can extrapolate at least modestly their performances under a warmer climate. Note that it is an often asked question for purely data-driven models like LSTM to lose accuracy if the drift is stronger. The issue is that there was a lack of a "substantially warmed" real dataset to assess this behavior. Previous benchmarks suggest the 15-year-scale trend to be accurately captured in the temporal test by LSTM, and more poorly captured for PUR (Feng et al., 2023). We have added these discussions in Section 3.3 of the revised manuscript to describe this finding.

[Figure]

*Figure R1. Boxplot of relative temperature differences between the test and training periods, calculated as (Test − Training) / Training. Each box represents the distribution of normalized temperature changes across basins for a specific meteorological forcing dataset: Daymet, NLDAS, and Maurer. Positive values indicate warming in the test period relative to the training period.*

*3. Table 2: What would happen if you were to train HBV using only the same 531-basin subset as for the LSTM instead of the 671 basins?*

Thanks for the question. We tested this by conducting the temporal experiments using only the 531-basin subset for the δHBV model. The results were largely similar to those obtained using all 671 basins shown in Figures R2-R3, indicating that the impact of reducing the training set size is limited in this context. We have incorporated some discussions about this in the revised manuscript.

We believe that training the δHBV model on the full 671-basin dataset is still beneficial, as the physical constraints inherent in the model allow it to make more effective use of available data, even when data quality is somewhat limited. That said, the added value from including the additional 140 basins appears to be marginal, and the choice of training on 531 versus 671 basins does not substantially affect the overall model performance.

[Figure]

*Figure R2. Comparison of δHBV simulations trained on 671 versus 531 basins across performance metrics (a)–(g), with test cases distinguished by varying x-axis labels.*

[Figure]

*Figure R3. Comparison of NSE spatial distributions for δHBV models trained on 671 versus 531 basins across different meteorological forcing datasets.*

*4. It could be useful to also provide similar plots to Fig. 3 in the appendix where HBV2 and HBV3 are used instead of HBV1.*

Thanks for the suggestions. We have plotted Figures R4 and R5 based on δHBV2 and δHBV3, and they show similar results, consistent with the conclusions. We have revised the manuscript accordingly.

[Figure]

*Figure R4. Scatter plots comparing the performance differences between hydrological models for the basins where LSTM outperformed δHBV (the basins where δHBV outperformed are not shown in this plot). The x-axis represents the NSE differences between $LSTM^2$ and $δHBV^2$ ($LSTM^2$ - $δHBV^2$), while the y-axis shows the NSE differences between $δHBV^{123}$ and $δHBV^2$ ($δHBV^{123}$- $δHBV^2$). Points are color-coded according to the NSE values of $δHBV^2$. The correlation coefficient (CORR) and p values between the x-axis values and the y-axis values, along with the median NSE value of $δHBV^2$ ($NSE_{med}$) on these basins, are also noted.*

[Figure]

*Figure R5. Scatter plots comparing the performance differences between hydrological models for the basins where LSTM outperformed δHBV (the basins where δHBV outperformed are not shown in this plot). The x-axis represents the NSE differences between $LSTM^3$ and $δHBV^3$ ($LSTM^3$ - $δHBV^3$), while the y-axis shows the NSE differences between $δHBV^{123}$ and $δHBV^3$ ($δHBV^{123}$- $δHBV^3$). Points are color-coded according to the NSE values of $δHBV^3$. The correlation coefficient (CORR) and p values between the x-axis values and the y-axis values, along with the median NSE value of $δHBV^3$ ($NSE_{med}$) on these basins, are also noted.*

*5. Fig. 4 and L439-L454: Can you expand on potential reasons for the lower model-skill in midwestern and western basins? Is human management of streamflow an important factor here, despite being CAMELS basins?*

Thanks for the suggestions. We have directly compared the spatial patterns of performance with the spatial distributions of evaporation, the dam number, and water use. We found that over the midwestern and western CONUS, there are also high evaporation climate conditions (Figure 2 of (Heidari et al., 2020), shown here as Figure R6) and a large number of dams (Figure 1 of (Ryan Bellmore et al., 2017), shown here as Figure R7), which tend to have complex water use processes that cannot be simulated via the models (Figure 11 of (Wada et al., 2016), shown here as Figure R8). All these factors indicate that anthropogenic influence can be an important factor that causes the model to perform poorly. And these factors have been implicitly expressed in lines 486-490 in Section 3.4 of the original manuscript. Based on your suggestions, these sentences have been revised to be clearer.

[Figure]

*Figure R6. Maps of current (a) aridity index and (b) evaporative index for the baseline period (1986–2015) from (Heidari et al., 2020).*

[Figure]

*Figure R7. Distribution of dams in the contiguous U.S. from the corrected figure of (Ryan Bellmore et al., 2017).*

[Figure]

Water Stress Index [demand / availability; dimensionless]

0 - 0.1    0.1 - 0.2    0.2 - 0.4    0.4 - 0.6    0.6 - 0.8    > 0.8

*Figure R8. Water Stress Index for 2010 calculated at a 6 min spatial resolution from (Wada et al., 2016)*

*6. Fig. 5: Please clarify. L399-L401 says there is a small difference when using 3 or 10 seeds, but Fig. 5 shows the difference between individual seeds and using 10 seeds. It is interesting to note in Fig. 5 that LSTM^multi with 10 seeds achieves a similar skill as (LSTM + HBV)^123 at least for the temporal test. This could also be an important conclusion.*

Thanks for the comments. It is true that LSTM, after ensembling different random seeds, increases much higher compared with the individual component and other ensemble strategies. But here, we want to show that there are smaller differences between ensembling 3 random seeds and ensembling 10 seeds. Specific metric values of the LSTM model with different random seeds can be found in Table D4 and Table D5 (0.81992 (3 seeds) v.s. 0.82425 (10 seeds)). We also implicitly describe the boost performance improvement of $LSTM_{seed}^{multi}$ and owe it to the instability of LSTM simulations in the original manuscript as, "The performance of $(LSTM + \delta HBV)^{123}$ ensemble proved more robust than $LSTM^{multi}$, with only a slight boost when we incorporated random seeds, i.e., $(LSTM + \delta HBV)_{seed}^{123}$. " in lines 390-391. Based on your suggestions, we have added some sentences to make it clearer in Section 3.1 of the revised

manuscript, "For LSTMs alone, different random seeds displayed higher variation, and ensembling them led to greater improvement than ensembling $(LSTM + \delta HBV)^{123}$ with additional random seeds. "

*7. Fig. 6: Suggestion to have LSTM models with shades of one color and HBV models with shades of a different color to better highlight the differences between LSTM and HBV mentioned in L461-L464.*

Thanks for the suggestions. We replotted the figure accordingly as shown in Figure R9 here.

[Figure]

*Figure R9. Comparisons between multi-basin-averaged streamflow observations and simulations across 531 basins. The time series points are displayed at four-day intervals for clarity and conciseness. Ensemble members based on the same model (LSTM or δHBV) but driven by different forcing datasets are shown in the same color to highlight the differences between models more clearly.*

*8. L467-L469: Suggestion to extend the sentence to clarify what is meant here.*

Thanks for the suggestion. We have revised the original sentence

"This highlights the critical importance of comprehensive training for each ensemble member to enable the development of distinct characteristics in their streamflow simulations, ultimately enhancing ensemble performance." in Section 3.2 of the original manuscript as,

"This highlights the critical importance of comprehensive training for each ensemble member, including diverse forcing inputs, full-period model calibration, and rigorous hyperparameter tuning, to ensure that each member develops distinct simulation behaviors. These differences allow the ensemble to better represent a range of hydrological responses, particularly under extreme or uncertain conditions. By capturing complementary strengths and compensating for individual weaknesses, such well-trained ensemble members collectively enhance the robustness and accuracy of streamflow simulations."

*Minor comments and technical corrections*

*1. L13: Replace "while" for "however".*

Thanks, revised.

*2. L20: Suggest deleting "utilized in two ways". Here it just raises the question which two ways? Also in L526 it would be good to explicitly mention the "two ways".*

Thanks for the suggestions. We have deleted the "utilized in two ways" in the Abstract section.
As for "*Also in L526 it would be good to explicitly mention the "two ways".*" in the conclusion section, we have revised the original words as,
"Three meteorological forcing datasets (Daymet, NLDAS, and Maurer) are employed to fully capture the characteristics of the two models. Their applications are also tested in two distinct ways: (1) by feeding all diverse forcing datasets simultaneously into a single LSTM model, and (2) by ensembling the outputs of multiple LSTM models, each trained separately using a single forcing dataset."

*3. L177-L179: Are all modifications to HBV1.0 of similar importance? Or can it be said which of them are more important?*

Thanks for the insightful comment. We are very cautious to make modifications, and to determine these modifications, we have evaluated various structural changes across multiple studies using diverse datasets. Each modification targets specific aspects of model improvement, and most contribute significantly to overall performance.

To address high-flow simulation challenges, we implemented three key modifications: the use of three dynamic parameters ($\gamma, \beta, \theta_{K0}$) during training and testing periods; the removal of log-transform normalization for precipitation; and the adoption of the normalized squared-error loss function.

Our recent study (Song et al., 2025b) shows that δHBV1.1p with three dynamic parameters ($\gamma, \beta, k_0$) outperforms the two-parameter version ($\gamma, \beta$). The dynamic shape coefficient ($\beta$) and evapotranspiration coefficient ($\gamma$) capture the nonlinear relationships between surface soil moisture and effective rainfall, as well as evapotranspiration. The dynamic $k_0$ parameter reflects variable water release rates influenced by changing groundwater levels, bank and wetland storage, and other factors. By remaining small during

low-flow periods and increasing during peak-flow events, dynamic $k_0$ allows the upper soil layer to retain more moisture before extreme events, thereby enhancing peak-flow contributions..

The elimination of log-transform normalization for precipitation, paired with the adoption of the normalized squared-error (NSE) loss function, synergistically enhances model performance. By removing log-transform normalization, the model becomes more sensitive to high precipitation events, thus better capturing high-flow conditions. Simultaneously, the normalized squared-error loss function amplifies the impact of significant deviations in peak flows, further improving the model's ability to predict high-flow events effectively (Frame et al., 2022; Kratzert et al., 2021; Song et al., 2025a, b; Wilbrand et al., 2023).

In contrast, maintaining dynamic parameters during warm-up periods offers marginal benefits while increasing computational costs. However, it provides a more realistic representation and mitigates potential uncertainties from initial conditions.

Following the feedback, we have revised and expanded the descriptions in Section 2.3 to enhance clarity.

*4. Table 2: Isn't there more recent data for PUB and PUR? Why are they trained only until 1999?*

Thanks for the question. All three tests (temporal, PUB, and PUR) are conducted using the same underlying dataset. However, due to differences in testing strategies, the computational cost for PUB and PUR is significantly higher than for the temporal test. Specifically, each complete evaluation requires 10 runs for PUB and 7 runs for PUR. Based on prior studies (Feng et al., 2021, 2023; Kratzert et al., 2019) and to balance computational efficiency with the objectives of our analysis, we limited the training period to data up to 1999. This choice allows us to preserve the core evaluation goals while keeping the computational demand manageable.

*5. Figure B1. Xlabel on right panels should be temperature (C), correct?*

Thanks for pointing it out. It has been replotted (Figure R10) and has been added in the revised manuscript.

[Figure]

*Figure R10. Probability density distributions of precipitation and temperature across three meteorological forcing datasets.*

*Reviewer #2*
*The manuscript presents an innovative ensemble strategy that combines a differentiable process-based model (δHBV) with a data-driven Long Short-Term Memory (LSTM) model, further diversified through the application of multiple meteorological forcing datasets. The approach is evaluated across a wide range of generalization scenarios (temporal extrapolation, PUB, and PUR) using the CAMELS dataset. Although the paper is well-written and the main ideas are clearly communicated, it would benefit from additional details in the methods and a deeper discussion of model complementarity and limitations.*

Thanks for the positive comments and constructive suggestions. We have revised the manuscript accordingly. Please find our point-by-point responses below.

*Strength*

*The proposed ensemble framework is conceptually strong and offers a well-justified combination of complementary data and algorithmic modeling paradigms.*
*The study is well evaluated across well-defined training protocols and temporal-spatial splits, which improves confidence in its generalizability.*
*The use of multiple data sources for meteorological forcings addresses input uncertainty better than traditional single-source modeling.*
*The results, specifically the finding that δHBV improves spatial generalization, have clear implications for prediction in ungauged regions.*

Thanks for the positive comments.

*Weakness*

*Interpretability: While the δHBV model's performance is shown to be beneficial in spatial generalization, the underlying reasons for this complementarity (e.g., structural constraints, parameter smoothness) are not deeply explored. A discussion of how each model contributes to ensemble diversity would strengthen the scientific value of the work.*

We thank the reviewer for this insightful comment regarding model interpretability and the complementarity between LSTM and δHBV within the ensemble framework. We fully agree that a deeper understanding of the relative contributions of each model would enhance the scientific value of our study. Besides this, we have also dug deeper and examine cases where the errors of LSTM and δHBV cancel each other.

[Figure]

*Figure R11. Spread values of each model for LSTM, δHBV, and LSTM + δHBV due to different meteorological forcings and random seeds across temporal, PUB, and PUR tests.*

[Figure]

*Figure R12. Spatial distributions of model spread values increase from δHBV and LSTM to the LSTM+δHBV ensemble across temporal, PUB, and PUR tests.*

[Figure]

*Figure R13. Distributions of observation–simulation pairs from LSTM and δHBV models along the 1:1 line across temporal, PUB, and PUR tests. Percentages of pairs lying above the 1:1 line for both models are also indicated.*

Since the benefits of the different ensemble members to the deterministic precision have been displayed in the original manuscript, we have conducted additional analyses in terms of ensemble variability as suggested. Specifically, we use the spread values (Li et al., 2021; Reichle and Koster, 2003), which are widely adopted to quantify ensemble variability, to further explore model complementarity. The spread value is calculated as follows,

$$Spread = \sqrt{\frac{1}{n}\frac{1}{e}\sum_{i=1}^{n}\sum_{j=1}^{e}(S_{i,j} - \mu_{S,i})^2}$$

Where $n$ is the number of simulated days, $e$ is the number of ensemble members, and $S$ is the simulations of each ensemble member, $\mu_S$ indicates the average of values.

Figure R11 presents the boxplots of spread values for ensemble simulations using random seed variations with LSTM, δHBV, and the combined LSTM + δHBV, across the temporal, PUB, and PUR test settings. We observe that the overall spread increases from temporal to PUB and PUR tests, reflecting growing uncertainty. Notably, δHBV consistently exhibits lower spread values than LSTM across all tests, indicating its higher stability. This aligns with our prior discussion: δHBV tends to constrain the learnable function space, thus having lower variability and potentially higher bias. This difference stems from their structural characteristics—δHBV is governed by more rigid physical constraints, which limit unrealistic dynamics and enhance stability, while LSTM is more flexible and capable of capturing patterns that may not be explicitly represented in physical models, such as human influences or unmodeled processes. The combination of both models (LSTM + δHBV) yields greater spread values, indicating enhanced ensemble diversity. This suggests that the two models offer complementary strengths—LSTM contributes flexibility and capacity to represent data-driven nuances, while δHBV anchors the ensemble with physically constrained behavior.

Figure R12 illustrates the spatial distributions of spread increase resulting from incorporating LSTM and δHBV, respectively, and further supports our previous analysis. Incorporating LSTM leads to an increase in spread values across all basins, reflecting its higher variability. In contrast, the δHBV model, characterized by stronger physical constraints and generally lower variability, results in a decrease in spread values for many basins. However, δHBV still contributes to a spread increase in most northern basins and gradually leads to spread increases in a larger number of basins across the CONUS. This suggests notable differences in simulated streamflow behavior between LSTM and δHBV, largely attributable to their distinct model structures. Figure R13 reveals relatively limited differences between the streamflow behaviors simulated by LSTM and δHBV, with LSTM generally producing higher streamflow estimates than δHBV. A more systematic investigation of these differences would be valuable in future studies.

Following the reviewer's suggestion, we have incorporated these analyses and discussions about the ensemble spread in Section 3.3 of the revised manuscript.

*Robustness and Sensitivity Analysis: The paper lacks an explicit assessment of how ensemble performance responds to errors or biases in the forcing datasets or uncertainty in model parameters. Including even a limited robustness analysis would improve confidence in the ensemble's reliability. Additionally, the authors should consider running one or two experiments to understand whether changing the size of the lookback window (i.e., the number of historical timesteps) for the LSTMs impacted the overall performance of the ensemble.*

Thanks for the suggestions. Based on them, we conducted several experiments using temporal tests to demonstrate the robustness of ensemble benefits under various factors, including precipitation errors, parameter uncertainties in the δHBV model, and hyperparameter uncertainties in the LSTM model.

Regarding sensitivity to the forcing datasets, we ran the δHBV and LSTM models under a temporal test, both without and with a precipitation error introduced by multiplying precipitation by 0.1, to examine differences across ensemble groups. The results, shown in Figures R14 and R15, indicate that although the performance of both LSTM and δHBV decreases when the precipitation error is introduced, the decrease is not substantial, demonstrating a certain degree of robustness to precipitation errors and some capacity of both models to adapt to such errors. Interestingly, LSTM and δHBV respond differently to this type of precipitation error: for LSTM, the error tends to reduce ensemble performance mainly under low and high flow regimes, whereas for δHBV, the reduction is more pronounced under low and middle flow regimes. These differences reflect the fact that LSTM does not need to respect mass balance and can adjust precipitation up or down internally, but has trouble learning the contrast, while δHBV needs to distort the low flow to capture the high flows. Despite these differences, the ensemble benefits remain significant and robust when comparing different ensemble groups and assessing the impact of precipitation errors.

[Figure]

*Figure R14. Simulation performance under the temporal test using the LSTM model with and without a 10% precipitation error (precipitation × 1.1), compared across metrics (a)–(g).*

[Figure]

*Figure R15. Simulation performance under the temporal test using the δHBV model with and without a 10% precipitation error (precipitation × 1.1), compared across metrics (a)–(g).*

Similar results are observed in cases investigating the effects of parameter uncertainties in δHBV (Figure R16) and hyperparameter uncertainties in LSTM (Figure R17). Regarding parameter uncertainties, we additionally ran a case using the δHBV model with fewer dynamic parameters—reducing the number from three in the benchmark case to two—by fixing the infiltration rate parameter $k_0$ as static to assess the resulting performance changes, which may reduce δHBV's ability to represent dynamic water release processes influenced by changing groundwater levels, bank and wetland storages, and other factors (Song et al., 2025b). This leads to increased structural errors and decreased model performance. Nevertheless,

the contribution of δHBV to ensemble simulations remains robust, with ensemble benefits substantially outweighing the negative effects of parameter uncertainties.

[Figure]

*Figure R16. Simulation performance under the temporal test using the δHBV model with 3 and 2 dynamic parameters, compared across metrics (a)–(g).*

Regarding hyperparameter uncertainties in the LSTM model, we focus on a key hyperparameter: the lookback window size, as suggested. We treat this parameter as having physical significance related to the temporal period rather than a typical hyperparameter. Therefore, we fix the window size to one year (365 days) to capture a full annual cycle while accounting for interannual variability. To evaluate the impact of

different window lengths, we include two additional scenarios with 182 and 730 timesteps. As shown in Figure R17, the LSTM model with a 365-day window generally achieves better performance across most scenarios. However, compared to the overall benefits of the ensemble, this difference is not substantial, indicating the robustness of ensemble simulations to variations in this LSTM hyperparameter.

[Figure]

*Figure R17. Simulation performance under the temporal test using the δHBV model on the time steps of 365, 182, and 730, compared across metrics (a)–(g).*

Although it is practically impossible to test the effects of all possible configurations on ensemble benefits, we expect these benefits to remain robust against other factors to some extent, based on the representative results presented. Following the suggestions, we have included these additional cases in Section 3.3 of the revised manuscript to further demonstrate the reliability of our ensemble simulations.

*Scalability and Practical Deployment: The manuscript does not address the computational or operational feasibility of deploying this ensemble framework in practice, especially over large domains or in real-time forecasting contexts. A short discussion (1-2 sentences) on this topic would add practical relevance.*

We appreciate the reviewer's suggestion to further discuss the computational and operational feasibility of deploying the ensemble framework. This point is partially addressed in Section 3.4 of the original manuscript, where we note:
"Moreover, ensemble simulations may face challenges when computational resources are limited and calculations are performed sequentially. However, we remain optimistic about these challenges, as the processes can be addressed by leveraging parallel computing with multiple GPUs, benefiting from ongoing advancements in computational power."
In response to the reviewer's comment, we have expanded this discussion for greater clarity, as follows:
"Ensemble simulations may face challenges when computational resources are constrained, particularly for large-scale or real-time applications. Nevertheless, we remain optimistic about overcoming these challenges due to several promising solutions. These include tailoring the hydrological model by simplifying less relevant components to specific simulation objectives (Clark et al., 2015; Kraft et al., 2022) and cloud-based computing infrastructures that offer scalable, on-demand resource allocation (He et al., 2024; Leube et al., 2013). Importantly, the majority of computational costs are incurred during model training. In practice, ensemble members are typically pre-trained by different research or application groups (Bodnar et al., 2025; Nearing et al., 2024; Song et al., 2025a), enabling direct reuse of these well-trained models and significantly improving computational efficiency."

**References:**

[revised manuscript text omitted]

---

## Referee Report (RR1)

**Review of "Ensembling Differentiable Process-based and Data-driven Models with Meteorological Forcing Datasets to Advance Streamflow Simulation"**

The authors have comprehensively answered the points raised during the previous review round, and incorporated changes that improved the manuscript. I consider that it can be published in its current state. Nevertheless, the following minor comments may contribute to further improve the final paper. The line numbers correspond to those in the file with highlighted changes.

L182: Rephrase. Suggested alternative: "Three additional modifications are included to address high-flow simulation challenges:…"

Section 2.5: I just wondered if the performance metrics from the different ensembles vary much depending on the season of the year. However, I realize that this might be out of the scope of the current study.

Fig. 5: If I understand it correctly, the plot compares single seeds (dashed translucent lines) versus the case when using 10 seeds (solid lines), while no lines show the results when averaging 3 seeds. However, in the text (L416-420) it is discussed as comparing the average 3 seeds versus the average of 10 seeds. It would be helpful to refer to in the text which are the NSE values obtained by averaging 3 seeds that are to be compared with the values shown in Fig. 5 obtained by averaging 10 seeds. Alternatively, consider adding lines to Fig. 5 corresponding to the average of 3 seeds.

L546: It seems it should be cited as "Bellmore et al., 2017": Bellmore, J.R., Duda, J.J., Craig, L.S., Greene, S.L., Torgersen, C.E., Collins, M.J., et al. (2017) Status and trends of dam removal research in the United States. *Wiley Interdisciplinary Reviews: Water*, **4**(2), e1164. Available from: https://doi.org/10.1002/wat2.1164

Fig. B1: Clarify in the caption what is the difference between the top and bottom panels.